# Domain Adaptation with Cauchy-Schwarz Divergence

**Wenzhe Yin**[1]     **Shujian Yu**[*2,4]     **Yicong Lin**[2]     **Jie Liu**[1]     **Jan-Jakob Sonke**[3]     **Efstratios Gavves**[1]

[1]University of Amsterdam, Amsterdam, The Netherlands
[2]Vrije Universiteit Amsterdam, Amsterdam, The Netherlands
[3]Netherlands Cancer Institute, Amsterdam, The Netherlands
[4]UiT - The Arctic University of Norway, Tromsø, Norway

## Abstract

Domain adaptation aims to use training data from one or multiple source domains to learn a hypothesis that can be generalized to a different, but related, target domain. As such, having a reliable measure for evaluating the discrepancy of both marginal and conditional distributions is crucial. We introduce Cauchy-Schwarz (CS) divergence to the problem of unsupervised domain adaptation (UDA). The CS divergence offers a theoretically tighter generalization error bound than the popular Kullback-Leibler divergence. This holds for the general case of supervised learning, including multi-class classification and regression. Furthermore, we illustrate that the CS divergence enables a simple estimator on the discrepancy of both marginal and conditional distributions between source and target domains in the representation space, without requiring any distributional assumptions. We provide multiple examples to illustrate how the CS divergence can be conveniently used in both distance metric- or adversarial training-based UDA frameworks, resulting in compelling performance. The code of our paper is available at `https://github.com/ywzcode/CS-adv`.

## 1 INTRODUCTION

Deep learning has achieved outstanding performance in different vision tasks, including image classification [He et al., 2016] and semantic segmentation [Ronneberger et al., 2015]. Typically, it is assumed that the training and test data are drawn from the same distribution. In reality, this assumption is often violated due to a variety of factors, such as changes in lighting conditions, viewpoints, and the appearance of objects. This discrepancy between the source

---

*Corresponding author (yusj9011@gmail.com).

and target domains is referred to as domain shift [Mansour et al., 2008, Yosinski et al., 2014], which can significantly degrade the generalization capability of the learned model.

Domain adaptation aims to mitigate the effects of domain shift by leveraging the knowledge acquired from one or multiple source domains to improve the model's performance in different, but related, target domains. Most of the previous methods aim to learn a domain-invariant feature representation $\mathbf{z}$ that has the same marginal distribution $p(\mathbf{z})$ across domains. Usually, this is achieved by either using different kinds of divergence measures, such as Maximum Mean Discrepancy (MMD) [Gretton et al., 2012, Zhang and Wu, 2020], Kullback-Leibler (KL) divergence [Nguyen et al., 2021b], Wasserstein distance which arises from the idea of optimal transport [Damodaran et al., 2018, Fatras et al., 2021], or adopting advanced optimization strategies such as adversarial training [Ganin et al., 2016, Long et al., 2015, Saito et al., 2018, Zhang et al., 2019, Du et al., 2021]. However, these approaches implicitly assume that the conditional distribution $p(y|\mathbf{z})$ remains the same, which is generally an overly optimistic assumption (cf. Fig. 1 in [Zhao et al., 2019]).

Moreover, due to the high dimensionality and continuous nature of representation $\mathbf{z}$, estimating the discrepancy of $p(y|\mathbf{z})$ in two domains is challenging. The above-mentioned distance measures, including MMD, KL divergence, and Wasserstein distance, have been considered to model the discrepancy of $p(y|\mathbf{z})$. However, earlier approaches either resort to matching $p(\mathbf{z}|y)$ instead (e.g., [Luo and Ren, 2021, Zhang and Wu, 2020]), or naïvely assume that such discrepancy is sufficiently small [Nguyen et al., 2021b]. Note that, given that $p(\mathbf{z})$ is aligned, aligning $p(y|\mathbf{z})$ by solely matching $p(\mathbf{z}|y)$ implicitly assumes that $p(y)$ is invariant.

The above issue motivates us to introduce the Cauchy-Schwarz (CS) divergence [Principe et al., 2000b,a] to the problem of unsupervised domain adaptation (UDA), in which the target domain is unlabeled. Firstly, we demonstrate that CS divergence can explicitly align the discrepancy

of $p(y|\mathbf{z})$ between source and target domains. Secondly, utilizing the CS divergence, we establish a tighter generalization bound in comparison to the commonly adopted KL divergence. This offers a theoretical guarantee on the improved performance of our overall model compared to other state-of-the-art (SOTA) approaches.

Our contributions can be summarized as follows:

1. To the best of our knowledge, this is the first attempt to introduce CS divergence to UDA for aligning $p(y|\mathbf{z})$.

2. We show that the CS divergence enables a tighter generalization bound on UDA than the popular KL divergence. Unlike classic generalization error bound [Ben-David et al., 2010] that only applies to the binary classification setup and is hard to optimize in practice, our bound applies to general domain adaptation tasks including multi-class classification and regression.

3. We provide a simple, non-parametric approach of estimating the CS divergence for both $p(\mathbf{z})$ and $p(y|\mathbf{z})$ between source and target domains, without relying on any distributional assumptions.

4. We show that the proposed CS divergence can be conveniently used in both distance metric- or adversarial training-based UDA frameworks. The CS divergence can also be smoothly integrated as a flexible plug-in module to improve modern UDA approaches.

## 2 RELATED WORK

**Domain Adaptation**    Prior research in the field has predominantly concentrated on aligning the marginal distribution $p(\mathbf{z})$ with a valid distance metric or in an adversarial training manner.

The most popular distance metric is Mean Maximum Discrepancy (MMD) [Gretton et al., 2012], which has been widely used in domain adaptation task [Pan et al., 2010, Ding et al., 2018, Long et al., 2015]. Similar to MMD, CORAL [Sun and Saenko, 2016] matches the first two moments of distributions. Other distance metrics, such as Wasserstein distance [Shen et al., 2018], manifold matching [Wang et al., 2018], optimal transport [Courty et al., 2017b], and margin disparity discrepancy (MDD) [Zhang et al., 2019] have also been used in matching the marginal distributions. Another line of methods utilizes the adversarial training [Ganin et al., 2016, Saito et al., 2018] for matching $p(\mathbf{z})$ through a min-max optimization.

However, such methods only consider matching marginal distributions. Aligning the conditional distributions of the source ($p^s(y|\mathbf{z})$) and target ($p^t(y|\mathbf{z})$) domains presents a considerable challenge due to the continuous and high-dimensional nature of $\mathbf{z}$, and the fact that the ground truth $y$ in the target domain is unknown in UDA setting. To address this issue, several attempts such as class condition

MMD [Zhang and Wu, 2020, Ge et al., 2023] and conditional kernel Brues (CKB) metric [Zhang and Wu, 2020] have been developed. It is important to note that the term "conditional" here refers to matching $p(\mathbf{z}|y) = \sum p(y = c_i)p(\mathbf{z}|y = c_i)$. Such formulation has two major limitations: 1) it implicitly assumes $p(y)$ is invariant (see Section 3.2 for a detailed discussion); 2) the scalability could be a problem when the number of classes is large, e.g., more than $1,000$ classes as commonly seen in vision tasks. Optimal transport has been used to match the joint distribution $p(\mathbf{z}, y)$ [Courty et al., 2017a, Damodaran et al., 2018, Fatras et al., 2021]. Typically, the transportation cost is represented as a weighted combination of costs in both feature and label spaces. In contrast, we match $p(\mathbf{z}, y)$ by explicitly modeling both $p(\mathbf{z})$ and $p(y|\mathbf{z})$, following the decomposition $p(\mathbf{z}, y) = p(y|\mathbf{z})p(\mathbf{z})$.

**Generalization Error Bound**    A tight generalization error bound coupled with a valid discrepancy measure plays a fundamental role in designing modern UDA approaches. Early studies have explored generalization bounds for UDA on binary classification with the aid of $\mathcal{H}\triangle\mathcal{H}$-divergence [Ben-David et al., 2010, Mansour et al., 2009]. Later, [Cortes and Mohri, 2011] extend the result to regression scenario, [Medina, 2015, Mohri and Medina, 2012] provide a tighter bound in on-line learning by introducing the $\mathcal{Y}$-discrepancy. [Cortes et al., 2019] use discrepancy minimization algorithm and solve a semi-definite programming (SDP) problem. Recently, [Acuna et al., 2021] refine the previous bounds and generalize them to a multi-class classification setting with the $f$-divergence, whereas [Richard et al., 2021] consider multi-source domain adaptation for regression with hypothesis-discrepancy.

**Cauchy-Schwarz Divergence**    Motivated by the well-known Cauchy-Schwarz (CS) inequality for square-integrable functions:

$$\left( \int p(\mathbf{x})q(\mathbf{x})d\mathbf{x} \right)^2 \leq \int p(\mathbf{x})^2 d\mathbf{x} \int q(\mathbf{x})^2 d\mathbf{x}, \quad (1)$$

with equality if and only if $p(\mathbf{x})$ and $q(\mathbf{x})$ are linearly dependent, the CS divergence [Principe et al., 2000a,b] defines the distance between probability density functions by measuring the tightness (or gap) of the left-hand side and right-hand side of Eq. (1) using the logarithm of their ratio:

$$D_{\text{CS}}(p; q) = -\log \left( \frac{(\int p(\mathbf{x})q(\mathbf{x})d\mathbf{x})^2}{\int p(\mathbf{x})^2 d\mathbf{x} \int q(\mathbf{x})^2 d\mathbf{x}} \right). \quad (2)$$

Eq. (1) also applies for two conditional distributions $p(y|\mathbf{x})$ and $q(y|\mathbf{x})$, the resulting conditional Cauchy-Schwarz

(CCS) divergence can be defined as [Yu et al., 2023]:

$$D_{\text{CS}}(p(y|\mathbf{x}); q(y|\mathbf{x})) = -2\log(\iint_{\mathcal{X},\mathcal{Y}} p(y|\mathbf{x})q(y|\mathbf{x})d\mathbf{x}dy)$$

$$+ \log(\iint_{\mathcal{X},\mathcal{Y}} p^2(y|\mathbf{x})d\mathbf{x}dy) + \log(\iint_{\mathcal{X},\mathcal{Y}} q^2(y|\mathbf{x})d\mathbf{x}dy)$$

$$= -2\log(\iint_{\mathcal{X},\mathcal{Y}} \frac{p(\mathbf{x},y)q(\mathbf{x},y)}{p(\mathbf{x})q(\mathbf{x})}d\mathbf{x}dy)$$

$$+ \log(\iint_{\mathcal{X},\mathcal{Y}} \frac{p^2(\mathbf{x},y)}{p^2(\mathbf{x})}d\mathbf{x}dy) + \log(\iint_{\mathcal{X},\mathcal{Y}} \frac{q^2(\mathbf{x},y)}{q^2(\mathbf{x})}d\mathbf{x}dy).$$

$$(3)$$

So far, due to the favorable properties of the CS divergence (e.g., enjoying closed-form expression for mixture-of-Gaussians [Kampa et al., 2011]), it has been successfully applied to deep clustering [Trosten et al., 2021], disentangled representation learning [Tran et al., 2022], point-set registration [Giraldo et al., 2017], just to name a few. However, several questions remain unanswered. For example, how does the CS divergence relate to the MMD or the KL divergence? Can these relations contribute to improving domain adaptation and generalization? How such new notions can be applied to construct practical domain adaptation models that achieve superior performance? We shall answer these questions in this paper.

# 3 METHOD

## 3.1 PRELIMINARY KNOWLEDGE OF UDA

In this paper, we focus on the unsupervised domain adaptation (UDA) problem. In UDA, we assume that there are $M$ labeled samples $\mathcal{D}^s = (\mathbf{x}_i^s, y_i^s)_{i=1}^M$ from the source domain and $N$ unlabeled samples $\mathcal{D}^t = (\mathbf{x}_j^t)_{j=1}^N$ from the target domain. Here, $\mathbf{x} \in \mathbb{R}^{d_x}$ represents the $d_x$-dimensional samples observed, and $y \in \{1, ..., K\}$ is the label of $K$ classes. Further, we denote the probability density function for source and target domains by $p^s$ and $p^t$, respectively. The primary goal of UDA is to find a hypothesis function $h = g \circ f : \mathbf{x} \mapsto \mathbf{y}$ such that the risk on the target domain is minimized. Here, $f : \mathbf{x} \mapsto \mathbf{z}$ is the feature extractor that maps the observations into a latent space, where $\mathbf{z} \in \mathbb{R}^{d_z}$ is the feature representation with $d_z$ dimensions. $g : \mathbf{z} \mapsto y$ is the classifier. We denote the prediction by $\hat{y} = g(\mathbf{z})$.

Most of the previous methods aim to learn a domain invariant feature representation $p(\mathbf{z}|\mathbf{x})$ by the feature extractor $f$ under the assumption of $p^s(\mathbf{x}) \neq p^t(\mathbf{x})$. This means that the distribution of the target domain is shifted from the source domain. Hence, it is intuitive to align the distribution $p(\mathbf{z})$ by a divergence $D(p^s(\mathbf{z}); p^t(\mathbf{z}))$. However, this ignores the conditional shift $p(y|\mathbf{z})$, which involves the label shift and classifier adaptation. Note that in UDA, we do not have access to $y$ in the target domain. The common strategy is using the discrete pseudo label from the prediction for matching

the class conditional discrepancy $p(\mathbf{z}|y)$. However, due to the CCS divergence being able to handle the continuous variables, we use the prediction vector from the classifier $\hat{y} = g(\mathbf{z})$ as the target label which leads to $p^t(\hat{y}|\mathbf{z})$.

## 3.2 DOMAIN SHIFT GENERALIZATION BOUND

We proceed by reviewing a newly developed KL-guided bound [Nguyen et al., 2021b], which can be used for general scenarios (including multi-class classification and regression) and makes no assumptions about the labeling mechanism (can be probabilistic or deterministic). According to [Nguyen et al., 2021b], the loss $l_{\text{test}}$ in the test distribution (a.k.a., target domain) satisfies:

$$l_{\text{test}} = \mathbb{E}_{p^t(\mathbf{x},y)}[-\log \hat{p}(y|\mathbf{x})] \leq \mathbb{E}_{p^t(\mathbf{z},y)}[-\log \hat{p}(y|\mathbf{z})]$$

$$= \int -\log \hat{p}(y|\mathbf{z})p^s(\mathbf{z},y)d\mathbf{z}dy +$$

$$\int -\log \hat{p}(y|\mathbf{z})[p^t(\mathbf{z},y) - p^s(\mathbf{z},y)]d\mathbf{z}dy$$

$$= l_{\text{train}} + \int -\log \hat{p}(y|\mathbf{z})[p^t(\mathbf{z},y) - p^s(\mathbf{z},y)]d\mathbf{z}dy$$

$$\leq l_{\text{train}} + \frac{M}{2} \int |p^t(\mathbf{z},y) - p^s(\mathbf{z},y)|d\mathbf{z}dy$$

$$\leq l_{\text{train}} + \frac{M}{2} \sqrt{2\int p^t(\mathbf{z},y) \log \frac{p^t(\mathbf{z},y)}{p^s(\mathbf{z},y)}d\mathbf{z}dy}$$

$$= l_{\text{train}} + \frac{M}{\sqrt{2}} \sqrt{D_{\text{KL}}(p^t(\mathbf{z},y); p^s(\mathbf{z},y))}$$

$$= l_{\text{train}} + \frac{M}{\sqrt{2}} \sqrt{D_{\text{KL}}(p^t(\mathbf{z}); p^s(\mathbf{z})) + D_{\text{KL}}(p^t(y|\mathbf{z}); p^s(y|\mathbf{z}))},$$

$$(4)$$

in which $l_{\text{train}}$ is the loss in the source domain, and the fourth line assumes that $-\log \hat{p}(y|\mathbf{z})$ is upper bounded by a constant $M$[1], the fifth line uses the famed Pinsker's inequality [Pinsker, 1964], which states that the total variation (TV) distance $D_{\text{TV}} = \frac{1}{2}\int |p(\mathbf{x}) - q(\mathbf{x})|d\mathbf{x}$ is upper bounded by the KL divergence $D_{\text{KL}} = \int p(\mathbf{x})\log\left(\frac{p(\mathbf{x})}{q(\mathbf{x})}\right)$ in the form of $D_{\text{TV}} \leq \sqrt{\frac{1}{2}D_{\text{KL}}}$. The last line follows the chain rule.

Referring to the second-to-last line of Eq. (4), achieving small test errors necessitates matching the joint distribution $p(\mathbf{z}, y)$, rather than solely focusing on the marginal distribution $p(\mathbf{z})$. This result is also consistent with [Zhao et al., 2019] and [Nguyen et al., 2021a]. Our paper utilizes the chain rule $p(\mathbf{z}, y) = p(y|\mathbf{z})p(\mathbf{z})$, indicating alignments for both $p(\mathbf{z})$ and $p(y|\mathbf{z})$. By contrast, existing literature (e.g., [Ge et al., 2023, Luo and Ren, 2021, Zhang and Wu, 2020]) often employs an alternative decomposition

---

[1]In classification, we can enforce this condition easily by augmenting the output softmax of the classifier so that each class probability is always at least $\exp(-M)$ [Nguyen et al., 2021b]. For example, if we choose $M = 4$, then $\exp(-M) \approx 0.02$.

$p(\mathbf{z}, y) = p(\mathbf{z}|y)p(y)$ but aligns only the classical conditional distribution $p(\mathbf{z}|y)$, thereby overlooking the impact of shift of $p(y)$.

Before providing a possibly tighter generalization error bound than the above-mentioned KL-guided bound, we first establish the connection between the CS divergence with respect to the KL divergence and the TV distance.

We proceed our analysis with a Gaussian assumption, in which their connections are demonstrated in Propositions 1 and 2. Note that, the Gaussian assumption on the learned representations (of deep neural networks) is commonly used in vision tasks [He et al., 2015, Ioffe and Szegedy, 2015].

**Proposition 1.** *For any d-variate Gaussian distributions* $p \sim \mathcal{N}(\mu_1, \Sigma_1)$ *and* $q \sim \mathcal{N}(\mu_2, \Sigma_2)$, *where* $\Sigma_1$ *and* $\Sigma_2$ *are positive definite, we have:*

$$D_{\mathrm{CS}}(p; q) \le \min\{D_{\mathrm{KL}}(p; q), D_{\mathrm{KL}}(q; p)\}. \quad (5)$$

*Proof.* All proofs of this paper are available in Section A of the Appendix. □

**Proposition 2.** *Let* $\Phi$ *be the cumulative distribution function of a standard normal distribution. Let* $p \sim \mathcal{N}(\mu_1, \Sigma_1)$ *and* $q \sim \mathcal{N}(\mu_2, \Sigma_2)$ *be any d-dimensional Gaussian distributions. We have:*

$$D_{\mathrm{TV}} \le \sqrt{D_{\mathrm{CS}}}, \quad (6)$$

*if one of the following conditions is satisfied:*

1. $\Sigma_1 = \Sigma_2 = \Sigma$ *and* $1/2\sqrt{\delta^\top \Sigma^{-1}\delta} \ge 2\Phi(\|\Sigma^{-1/2}\delta\|_2/2) - 1$, *where* $\delta = \mu_1 - \mu_2$;

2. $\sum_{i=1}^d \log\left(\frac{2 + \lambda_i + 1/\lambda_i}{4}\right) \ge 4$, *where* $\lambda_i$ *is the i-th eigenvalue of* $\Sigma_2^{-1}\Sigma_1$.

The conditions above in Proposition 2 are easily met, especially when $p$ and $q$ are not sufficiently similar and the variable dimension $d$ is large. For example, when $d = 1024$ as in our ResNet50 feature exactor, it suffices to require $\frac{2 + \lambda_i + 1/\lambda_i}{4} \ge 1.003$, which implies $\sum_{i=1}^d \log\left(\frac{2 + \lambda_i + 1/\lambda_i}{4}\right) \ge 4$. Note that, both CS and KL divergences are unbounded, whereas the TV distance is confined by an upper limit of 1.

In fact, the above connections can be extended to general distributions without assuming Gaussianity, as demonstrated in Propositions 3 and 4, respectively.

**Proposition 3.** *For any density functions* $p : \mathbb{R}^d \to \mathbb{R}_{\ge 0}$ *and* $q : \mathbb{R}^d \to \mathbb{R}_{\ge 0}$, *let* $K$ *be an integration domain over which* $p$ *and* $q$ *are Riemann integrable. Suppose* $|K| < \infty$, *where* $|K|$ *denotes the volume. Then*

$$C_1\left[D_{\mathrm{CS}}(p; q) - \log|K| + 2\log C_2\right] \le D_{\mathrm{KL}}(p; q), \quad (7)$$

*where* $C_1 = \int_K p(\mathbf{x})\,\mathrm{d}\mathbf{x}$, $C_2 = C_1\left(\int_K p^2(\mathbf{x})\,\mathrm{d}\mathbf{x} \int_K q^2(\mathbf{x})\,\mathrm{d}\mathbf{x}\right)^{-1/4}$. *Clearly, for* $K$ *such that* $|K \cap S| \gg 0$, *where* $S = \{\mathbf{x} : p(\mathbf{x}) > 0\}$, *one can have* $C_1 \approx 1$.

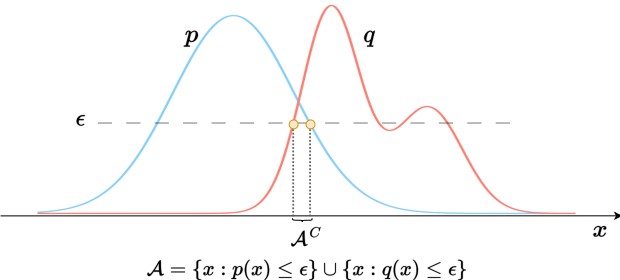

Figure 1: A graphical illustration of the sets $\mathcal{A}_\epsilon$ and $\mathcal{A}_\epsilon^\complement$ defined in Proposition 4.

**Proposition 4.** *For any density functions* $p$ *and* $q$, *and any* $\epsilon > 0$, *let* $\mathcal{A}_\epsilon = \{\mathbf{x} : p(\mathbf{x}) \le \epsilon\} \cup \{\mathbf{x} : q(\mathbf{x}) \le \epsilon\}$ *and* $\mathcal{A}_\epsilon^\complement$ *be its complement (see Fig. 1). Moreover, define* $T_{\mathcal{A}_\epsilon^\complement} = \sup\{p(\mathbf{x})q(\mathbf{x}), \mathbf{x} \in \mathcal{A}_\epsilon^\complement\}$ *and* $\left|\mathcal{A}_\epsilon^\complement\right|$ *to denote the "length" of the set* $\mathcal{A}_\epsilon^\complement$ *(strictly speaking, the Lebesgue measure of the set* $\mathcal{A}_\epsilon^\complement$). *Suppose there exists an* $\epsilon > 0$ *such that* $T_{\mathcal{A}_\epsilon^\complement}\left|\mathcal{A}_\epsilon^\complement\right| < \infty$ *and* $C_3 = \int p^2(\mathbf{x})\,\mathrm{d}\mathbf{x} \int q^2(\mathbf{x})\,\mathrm{d}\mathbf{x} \ge \exp(2)\left(2\epsilon + T_{\mathcal{A}_\epsilon^\complement}\left|\mathcal{A}_\epsilon^\complement\right|\right)^2$, *then*

$$D_{\mathrm{TV}}(p; q) \le \sqrt{D_{\mathrm{CS}}(p; q)}. \quad (8)$$

In our context, where $p$ and $q$ may differ substantially, using $D_{\mathrm{TV}}$ can be too restrictive to yield meaningful results. This is because $D_{\mathrm{TV}}$ measures the largest possible difference between $p$ and $q$, and it rapidly reaches its upper bound of 1, particularly when the location parameters of $p$ and $q$ differ. In this case, one is no longer be able to distinguish the distance between any sufficiently distinct pairs of $(p, q)$. Furthermore, as shown in Proposition 3, the inequality $C_1\left[D_{\mathrm{CS}}(p; q) - \log|K| + 2\log C_2\right] \le D_{\mathrm{KL}}(p; q)$ holds for any density functions $p$ and $q$, where $C_1 = \int_K p(\mathbf{x})\,\mathrm{d}\mathbf{x} > 0$ and $C_2 = C_1\left(\int_K p^2(\mathbf{x})\,\mathrm{d}\mathbf{x} \int_K q^2(\mathbf{x})\,\mathrm{d}\mathbf{x}\right)^{-1/4}$. Note that $C_1$ and $C_2$ are not conditions, but two constant values that depend on the distributions themselves. Furthermore, according to Proposition 4, if $p$ and $q$ do not sufficiently overlap, as ensured by the condition $\int p^2(\mathbf{x})\,\mathrm{d}\mathbf{x} \int q^2(\mathbf{x})\,\mathrm{d}\mathbf{x} \ge \exp(2)\left(2\epsilon + T_{\mathcal{A}_\epsilon^\complement}\left|\mathcal{A}_\epsilon^\complement\right|\right)^2$ for some $\epsilon > 0$, then $D_{\mathrm{TV}} \le \sqrt{D_{\mathrm{CS}}}$.

By combining these results, we have the following general

error bound:

$$l_{\text{test}} \le l_{\text{train}} + M D_{\text{TV}}(p^t(\mathbf{z}, y); p^s(\mathbf{z}, y))$$
$$\le l_{\text{train}} + M \sqrt{D_{\text{CS}}(p^t(\mathbf{z}, y); p^s(\mathbf{z}, y))}$$
$$\le l_{\text{train}} +$$
$$M \sqrt{C_1^{-1} D_{\text{KL}}(p^t(\mathbf{z}, y); p^s(\mathbf{z}, y)) + \log|K| - 2\log C_2},$$
(9)

if $\int p^2(\mathbf{x})\,d\mathbf{x} \int q^2(\mathbf{x})\,d\mathbf{x} \ge \exp(2)\left(2\epsilon + T_{\mathcal{A}_\epsilon^{\complement}}\left|\mathcal{A}_\epsilon^{\complement}\right|\right)^2$ for some $\epsilon > 0$. The latter condition is quite feasible to be satisfied. More discussion is in Remark 4 in the Appendix.

Similar to TV distance and most of $f$-divergence measures [Collet, 2019], the CS divergence does not satisfy the chain rule, indicating that the joint CS divergence cannot be expressed as the sum of marginal and conditional divergence. However, in practice, one can nevertheless control the joint divergence by minimizing the marginal and conditional counterparts separately. For simplicity, in this paper, we aim at minimizing:

$$l_{\text{train}} + M \sqrt{D_{\text{CS}}(p^t(\mathbf{z}); p^s(\mathbf{z})) + D_{\text{CS}}(p^t(y|\mathbf{z}); p^s(y|\mathbf{z}))}.$$
(10)

Although the KL-guided bound in Eq. (4) implies the necessity of minimizing the conditional divergence of $p(y|\mathbf{z})$, [Nguyen et al., 2021b] neglect this term (due to difficulty of estimation) and assume $p^t(y|\mathbf{z})$ and $p^s(y|\mathbf{z})$ are sufficiently close, which may not hold true [Zhao et al., 2020].

### 3.3 ESTIMATION OF CAUCHY-SCHWARZ DIVERGENCE

Suppose we have $M$ labeled samples $\{\mathbf{x}_i^s, y_i^s\}_{i=1}^M$ from the source domain and $N$ unlabeled samples $\{\mathbf{x}_i^t\}_{i=1}^N$ from the target domain, let us denote the predicted class probabilities for $\{\mathbf{x}_i^s\}_{i=1}^M$ and $\{\mathbf{x}_i^t\}_{i=1}^N$ are respectively $\{\hat{y}_i^s\}_{i=1}^M$ and $\{\hat{y}_i^t\}_{i=1}^N$, the following two propositions provide the empirical estimator of $D_{\text{CS}}(p^s(\mathbf{z}); p^t(\mathbf{z}))$ and $D_{\text{CCS}}(p^s(y|\mathbf{z}); p^t(y|\mathbf{z}))$. Moreover, in the following two remarks, we discuss the relationship and difference between (conditional) CS divergence and (conditional) MMD.

**Proposition 5** (Empirical Estimator of $D_{\text{CS}}(p^s(\mathbf{z}); p^t(\mathbf{z}))$ [Jenssen et al., 2006]). *Given extracted features from two domains $\{\mathbf{z}_i^s\}_{i=1}^M$ and $\{\mathbf{z}_i^t\}_{i=1}^N$, the empirical estimator of $D_{CS}(p^s(\mathbf{z}); p^t(\mathbf{z}))$ is given by:*

$$\widehat{D}_{CS}(p^s(\mathbf{z}); p^t(\mathbf{z})) = \log\left(\frac{1}{M^2}\sum_{i,j=1}^M \kappa(\mathbf{z}_i^s, \mathbf{z}_j^s)\right) +$$
$$\log\left(\frac{1}{N^2}\sum_{i,j=1}^N \kappa(\mathbf{z}_i^t, \mathbf{z}_j^t)\right) - 2\log\left(\frac{1}{MN}\sum_{i,j=1}^{M,N} \kappa(\mathbf{z}_i^s, \mathbf{z}_j^t)\right),$$
(11)

*where $\kappa$ is a kernel function such as Gaussian $\kappa_\sigma(\mathbf{z}, \mathbf{z}') = \exp(-\|\mathbf{z} - \mathbf{z}'\|_2^2/2\sigma^2)$.*

**Remark 1.** *The CS divergence is closely related to the MMD [Gretton et al., 2012]. In fact, the empirical estimator of the "biased" MMD can be expressed as:*

$$\widehat{MMD}^2(p^s; p^t) = \frac{1}{M^2}\sum_{i,j=1}^M \kappa(\mathbf{z}_i^s, \mathbf{z}_j^s)$$
$$+ \frac{1}{N^2}\sum_{i,j=1}^N \kappa(\mathbf{z}_i^t, \mathbf{z}_j^t) - \frac{2}{MN}\sum_{i=1}^M\sum_{j=1}^N \kappa(\mathbf{z}_i^s, \mathbf{x}_j^t).$$
(12)

*Comparing Eq. (11) with Eq. (12), we observe that the CS divergence estimator puts a "logarithm" on each term of that of MMD. Both estimators capture the within-distribution similarity subtracted by cross-distribution similarity, similar to the energy distance [Sejdinovic et al., 2013].*

**Proposition 6** (Empirical Estimator of $D_{\text{CCS}}(p^s(\hat{y}|\mathbf{z}); p^t(\hat{y}|\mathbf{z}))$ [Yu et al., 2023]). *Given features $\mathbf{z}$ and the corresponding predictions $\hat{y}$ from two domains, $\{\mathbf{z}_i^s, \hat{y}_i^s\}_{i=1}^M$ and $\{\mathbf{z}_i^t, \hat{y}_i^t\}_{i=1}^N$. Let $K^s$ and $L^s$ denote, respectively, the Gram matrices for the variable $\mathbf{z}$ and the predicted output $\hat{y}$ in the source distribution. Similarly, let $K^t$ and $L^t$ denote, respectively, the Gram matrices for the variable $\mathbf{z}$ and the predicted out $\hat{y}$ in the target distribution. Meanwhile, let $K^{st} \in \mathbb{R}^{M \times N}$ (i.e., $(K^{st})_{ij} = \kappa(\mathbf{z}_i^s - \mathbf{z}_j^t)$) denote the Gram matrix from source distribution to target distribution for input variable $\mathbf{z}$, and $L^{st} \in \mathbb{R}^{M \times N}$ the Gram matrix from source distribution to target distribution for predicted output $\hat{y}$. Similarly, let $K^{ts} \in \mathbb{R}^{N \times M}$ (i.e., $(K^{ts})_{ij} = \kappa(\mathbf{z}_i^t - \mathbf{z}_j^s)$) denote the Gram matrix from target distribution to source distribution for input variable $\mathbf{z}$, and $L^{ts} \in \mathbb{R}^{N \times M}$ the Gram matrix from target distribution to source distribution for predicted output $\hat{y}$. The empirical estimation of $D_{CCS}(p^s(\hat{y}|\mathbf{z}); p^t(\hat{y}|\mathbf{z}))$ is given by:*

$$\widehat{D}_{CCS}(p^s(\hat{y}|\mathbf{z}); p^t(\hat{y}|\mathbf{z}))$$
$$\approx \log\left(\sum_{j=1}^M \left(\frac{\sum_{i=1}^M K_{ji}^s L_{ji}^s}{(\sum_{i=1}^M K_{ji}^s)^2}\right)\right) + \log\left(\sum_{j=1}^N \left(\frac{\sum_{i=1}^N K_{ji}^t L_{ji}^t}{(\sum_{i=1}^N K_{ji}^t)^2}\right)\right)$$
$$- \log\left(\sum_{j=1}^M \left(\frac{\sum_{i=1}^N K_{ji}^{st} L_{ji}^{st}}{(\sum_{i=1}^M K_{ji}^s)(\sum_{i=1}^N K_{ji}^{st})}\right)\right)$$
$$- \log\left(\sum_{j=1}^N \left(\frac{\sum_{i=1}^M K_{ji}^{ts} L_{ji}^{ts}}{(\sum_{i=1}^M K_{ji}^{ts})(\sum_{i=1}^N K_{ji}^t)}\right)\right).$$
(13)

**Remark 2.** *Estimating the divergence between $p^s(\hat{y}|\mathbf{z})$ and $p^t(\hat{y}|\mathbf{z})$ is a non-trivial task. An alternative choice is the conditional MMD by [Ren et al., 2016]:*

$$\widehat{D}_{MMD}(p^s(\hat{y}|\mathbf{z}); p^t(\hat{y}|\mathbf{z})) = \text{tr}(K^s(\tilde{K}^s)^{-1} L^s(\tilde{K}^s)^{-1}) +$$
$$\text{tr}(K^t(\tilde{K}^t)^{-1} L^t(\tilde{K}^t)^{-1}) - 2\,\text{tr}(K^{st}(\tilde{K}^t)^{-1} L^{ts}(\tilde{K}^s)^{-1}),$$
(14)

in which tr *denotes the trace,* $\tilde{K} = K + \lambda I$. *Obviously, CS divergence avoids introducing a hyperparameter* $\lambda$ *and the necessity of matrix inverse, which improves computational efficiency and stability. See also experiments in Section 4.1.*

## 3.4 NETWORK TRAINING

We demonstrate how to use CS and conditional CS (CCS) divergences in both distance metric- and adversarial training-based UDA frameworks in a convenient way.

**Distance Metric Minimization** Given a neural network $h_\theta = f \circ g$, where $\mathbf{z} = f(\mathbf{x})$ is the learned features and $g : \mathbf{z} \mapsto y$ is a classifier. It is straightforward to use distance metrics to learn domain-invariant $f$ and $g$ (without introducing any new modules or training schemes). Specifically, the objective to train $h_\theta$ consists of the training loss on the source domain and a distribution discrepancy loss on both $p(\mathbf{z})$ and $p(y|\mathbf{z})$. For the former, we adopt the cross-entropy loss $L_{CE} = \frac{1}{M} \sum_{i=1}^{M} -y_i^s \log \hat{y}_i^s$. For the latter, we include both $D_{CS}(p^s(\mathbf{z}); p^t(\mathbf{z}))$ and $D_{CCS}(p^s(\hat{y}|\mathbf{z}); p^t(\hat{y}|\mathbf{z}))$, estimated with Eq. (11) and Eq. (13), respectively.

**Conditional Adversarial Training** We incorporate our CS and CCS divergences into a popular bi-classifier adversarial training framework [Saito et al., 2018] to attain SOTA performance. The bi-classifier adversarial training method utilizes two classifiers $g_1$ and $g_2$ as a discriminator. By maximizing the discrepancy between the two classifiers' output, the framework detects target samples that are outside the support of the source domain. Then, minimizing the discrepancy is for fooling the generator (feature extractor), which makes the features inside the support of the source with respect to the decision boundary.

As shown in Fig. 2, we model the alignment in two parts: 1) the minimization of $D_{CS}(p^s(\mathbf{z}); p^t(\mathbf{z})$ for learning domain-invariant representation; 2) the minimization of $D_{CCS}(p_1^t(\hat{y}|\mathbf{z}); p_2^t(\hat{y}|\mathbf{z}))$ for the conditional classifier adaptation adversarial training. We elaborate on the details of our training procedures by the following steps:

**Step 1** Learn feature extractor $f$ and two classifiers, $g_1$ and $g_2$, jointly by minimizing the classification loss $L_{cls}$ and the discrepancy loss $D_{CS} + D_{CCS}$ between the source domain and the target domain:

$$\min_{f, g_1, g_2} L_{cls} + \lambda D_{CS}(p^s(\mathbf{z}), p^t(\mathbf{z}))$$
$$+ \beta \sum_{n=1}^{2} D_{CCS}(p_n^s(\hat{y}|\mathbf{z}), p_n^t(\hat{y}|\mathbf{z})), \quad (15)$$

where $\lambda, \beta$ are weighting hyperparameters, $L_{cls}$ is the empirical risk over two classifiers in the source domain with additional entropy constraints in the target domain. Namely,

$$L_{cls} = \frac{1}{2} \sum_{n=1}^{2} (L_{CE}(g_n(f(\mathbf{x}^s)), y^s) + \gamma L_{Ent}(g_n(f(\mathbf{x}^t)))),$$
$$(16)$$

where $\gamma$ is a trade-off parameter, $L_{CE}$ is the cross-entropy loss, and $L_{Ent} = \frac{1}{N} \sum_{i=1}^{N} -\hat{y}_i^t \log \hat{y}_i^t$ ([Grandvalet and Bengio, 2004, Long et al., 2018, Luo and Ren, 2021, Du et al., 2021]) is a widely used constraint in domain adaptation.

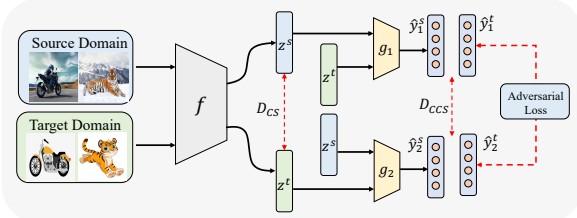

Figure 2: The framework of the proposed conditional bi-classifier adversarial learning method with CS and CCS divergences. Feature extractor $f$ is used to obtain representations $\mathbf{z}^s$ and $\mathbf{z}^t$ for the source and target domains, respectively. Two classifiers $g_1$ and $g_2$ are used as a discriminator. CS divergence directly minimizes the discrepancy of $p(\mathbf{z})$ between two domains. CCS divergence measures the disagreement between two classifiers (adversarial loss).

**Step 2** We fix parameters of the feature extractor $f$, and subsequently update the classifiers $g_1$ and $g_2$. By maximizing the conditional divergence between the two classifiers on the target domain, the discriminator (classifiers) is trained to identify the target samples that are outside the support of decision boundaries. To maintain classification accuracy on the source domain, classification loss is also used.

$$\min_{g_1, g_2} L_{cls} - D_{CCS}(p_1^t(\hat{y}|\mathbf{z}); p_2^t(\hat{y}|\mathbf{z})). \quad (17)$$

**Step 3** We fix the parameters of the two classifiers $g_1$ and $g_2$, and subsequently update the feature extractor $f$. The aim is to minimize the divergence between the probabilistic outputs of the two classifiers for training the feature exactor to fool the discriminator. This can be formalized as follows:

$$\min_{f} D_{CCS}(p_1^t(\hat{y}|\mathbf{z}); p_2^t(\hat{y}|\mathbf{z})). \quad (18)$$

## 4 EXPERIMENTS

We denote our distance metric-based approach as **CS+CCS** and adversarial training-based approach as **CS-adv**, and evaluate their performance on both synthetic and real-world datasets. Our experimental evaluation is divided into three parts: 1). advantages of CS divergence over other popular distance measures such as MMD and KL divergence in terms of statistical power (Sec. 4.1.1) and practical UDA performance (Sec. 4.1.2); 2). the superior performance of CS-adv with respect to other SOTA (Sec. 4.2); 3). the flexibility of CCS divergence as an injective module (Sec. 4.3).

**Datasets** We use three datasets in our experiments. 1) **Digits** The Digits dataset [Long et al., 2018] consists of two

parts, **MNIST** and **USPS**, which leads to two adaptation tasks, M→U and U→M. 2) **Office-Home** The Office-Home dataset [Venkateswara et al., 2017] has four domains: Art (Ar), Clipart (Cl), Product (Pr), and Real-World (Rw), which results in 12 domain adaptation tasks. The overall dataset has 15,500 images with 65 classes. 3) **Office-31** The Office-31 dataset [Saenko et al., 2010] has three domains: Amazon (A), Webcam (W), and DSLR (D), which results in 6 tasks. The overall Office-31 dataset contains 4,652 images with 31 categories. 4) **VisDA17** Additionally, we use a large scale real-world dataset VisDA17 [Peng et al., 2017] to have a fair comparison with KL [Nguyen et al., 2021b].

## 4.1 COMPARISONS AMONG (CONDITIONAL) CS DIVERGENCE, MMD AND KL DIVERGENCE

In this section, we first demonstrate that our conditional CS (CCS) divergence is statistically more powerful to discriminate two conditional distributions $p^s(y|\mathbf{x})$ and $p^t(y|\mathbf{x})$. Then, we compare our CS+CCS with both MMD-based approaches and the recently developed KL-based approach [Nguyen et al., 2021b]. Note that, all approaches are implemented without adversarial training, but only use different distance metrics to match $p(\mathbf{z})$ (or $p(y|\mathbf{z})$).

### 4.1.1 Statistical Test

For comparison purpose, we evaluate the performance of both conditional KL divergence estimated with the $k$-NN estimator [Wang et al., 2009] ($k = 3$) and the conditional MMD. In domain adaptation, $\text{MMD}(p^s(y|\mathbf{x}); p^t(y|\mathbf{x}))$ is rarely explicitly evaluated, due to the difficulty of estimation. Rather, a much more popular strategy is to evaluate the class conditional MMD (i.e., $\sum_{i=1}^{K} \text{MMD}(p^s(\mathbf{x}|y = c_i); p^t(\mathbf{x}|y = c_i))$ and $c_i$ indicates the $i$-th class). For the sake of comprehensiveness, we test the performances of both strategies and measure $\text{MMD}(p^s(y|\mathbf{x}); p^t(y|\mathbf{x}))$ with the estimator in [Ren et al., 2016].

We follow [Zheng, 2000] and generate 3 sets of data that have distinct conditional distributions: (a) $t = 1 + \sum_{i=1}^{d} x_i + \epsilon$, where $d$ refers to the dimension of explanatory variable $\mathbf{x}$, $\epsilon$ denotes standard normal distribution; the labeling rule is $y = 1$ if $t \geq 0$, otherwise $y = 0$. (b) $t = 1 + \sum_{i=1}^{d} \log(x_i) + \epsilon$; the labelling rule is again $y = 1$ if $t \geq 0$, otherwise $y = 0$. (c) $t = 1 + \sum_{i=1}^{d} x_i + \epsilon$; the labelling rule becomes $y = 1$ if $t \geq 1$, otherwise $y = 0$. For each set, the input $\mathbf{x}$ is fixed to be Gaussian, i.e., $p(\mathbf{x})$ remains the same, but $p(y|\mathbf{x})$ differs.

We generate 200 samples from each set and set $d = 10$. We apply a permutation test with significance level $\alpha = 0.05$ to test for the null hypothesis $H_0$ stating that two sets of data share a common conditional distribution, against the alternative hypothesis $H_1$ that suggests the conditional distributions are different. The results in Table 1 suggest

that our CCS is much more powerful. Details about the permutation test and data visualization are provided in the Appendix (Section C.1).

More specifically, by stating that a test statistic is more (statistically) powerful, we mean that it has a greater probability in finite samples of correctly rejecting the null hypothesis (i.e., two conditional distributions are equal) in favor of the alternative hypothesis (i.e., the two conditional distributions differ). That is, the statistic has a smaller Type-II error. Table 1 shows that our CCS method exhibits an empirical probability of 0.72 in correctly distinguishing between distribution (a) and distribution (c), while the probabilities associated with class CMMD and CMMD are notably lower at 0.20 and 0, respectively.

The main diagonal elements in the table refer to empirical Type-I error (i.e., the probability of falsely rejecting the null hypothesis). Given our chosen significance level of $\alpha = 0.05$, one would ideally expect the main diagonal elements to be close to 0.05. The results indicate that all methods exhibit similar performance in terms of size control. Hence, it can be concluded that the CCS approach offers statistically higher power with good size control.

|  | CCS | | | class CMMD | | | CMMD | | | CKL | | |
|---|---|---|---|---|---|---|---|---|---|---|---|---|
|  | (a) | (b) | (c) | (a) | (b) | (c) | (a) | (b) | (c) | (a) | (b) | (c) |
| (a) | 0.06 | 1 | 0.72 | 0.05 | 1 | 0.17 | 0.02 | 0 | 0.01 | 0.01 | 1 | 0.25 |
| (b) | 1 | 0.08 | 1 | 1 | 0.03 | 1 | 0 | 0.05 | 0 | 1 | 0.07 | 1 |
| (c) | 0.72 | 1 | 0.07 | 0.20 | 1 | 0.06 | 0 | 0 | 0.04 | 0.25 | 1 | 0.07 |

Table 1: Percent of rejecting $H_0$ hypothesis for conditional CS in Eq. (13), class conditional MMD, conditional MMD with estimator in [Ren et al., 2016], conditional KL with $k$-NN estimator [Wang et al., 2009]. An ideal result is a full-one matrix with $\alpha = 0.05$ on the main diagonal.

### 4.1.2 The Performance of CS+CCS

**Comparison with MMD** We demonstrate the advantages of CS and CCS over MMD in practical UDA tasks. To this end, we conduct an ablation study on the Digits **M→U** task. In this example, we match the marginal and conditional distributions with plain CS and CCS divergences without any adversarial training techniques. We use LeNet [LeCun et al., 1998] as the feature extractor and a nonlinear classifier with two fully connected layers and ReLU activation.

We compare CS, CCS, and CCS+CS divergences with MMD and joint probability MMD (JPMMD) [Zhang and Wu, 2020] that approximates $D(p^s(\mathbf{z}, y); p^t(\mathbf{z}, y))$ with $\mu_1 D(p^s(\mathbf{z}); p^t(\mathbf{z})) + \mu_2 D(p^s(\mathbf{z}|y); p^t(\mathbf{z}|y))$ (not $D(p^s(y|\mathbf{z}); p^t(y|\mathbf{z}))$). As shown in Fig. 3, all our CS divergence-based adaptations are consistently better than MMD and JPMMD. This means that our CCS divergence is better in modeling conditional alignment. Also, CCS has a better adaptation ability than CS divergence, while combining the CS and CCS divergences (CCS+CS) has the best

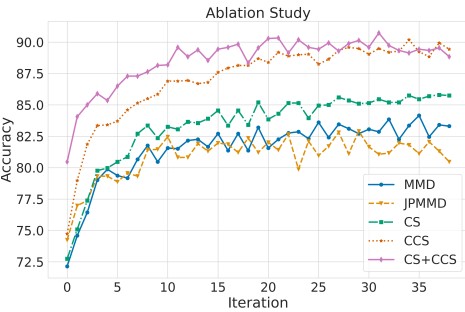

Figure 3: The ablation study of the **CS** and **CCS** components in MNIST to USPS task, comparing with MMD and joint distribution MMD (JPMMD).

| Method | M→U | U→M | VisDA17 |
|---|---|---|---|
| KL (Reproduced) | 98.2 | 97.2 | 52.3 |
| KL+CCS | **98.4** | **97.3** | **64.1** |
| CS+CCS | **98.3** | **97.9** | **64.5** |

Table 2: Results on Digits and VisDA17 datasets. KL+CCS integrates CCS divergence with KL. CS+CCS replaces KL with CS and CCS divergences.

performance. To have a better understanding of the learned representations on the two domains, we draw t-SNE [Van der Maaten and Hinton, 2008] visualization in Section C.4 in the Appendix. Additionally, we aim to add the comparison with conditional MMD in Eq. (14). However, the training fails due to the unstable numerical matrix inverse. This issue can also be observed in our statistical test in Table 1.

**Comparison with KL**    Furthermore, we draw a comparison with KL-based approach [Nguyen et al., 2021b]. We observed that its performance is very sensitive to network architecture choices. Moreover, with a nonlinear classifier, the model hardly converges. This point is also acknowledged by the authors[2]. The unstability of KL divergence can also be attributed to the term $\log(\frac{p(x)}{q(x)})$ which is likely to explode when $q(x) \to 0$. In contrast, our CS divergence can easily be adapted to different frameworks. To ensure fairness, we incorporated our CS and CCS divergences into [Nguyen et al., 2021b] framework, maintaining the identical architecture and hyperparameters. The comparative results are presented in Table 2. The proposed CS+CCS surpasses KL on both the Digits and VisDA17 datasets. Note that our reproduced results for KL on the VisDA17 dataset are lower than the reported scores. This may be attributed to the different computing environments and our adoption of a batch size of 128, as opposed to the original 256, due to memory limitations. Furthermore, we show that the performance of KL [Nguyen et al., 2021b] can be further improved by integrating our CCS regularization (KL+CCS). This result indicates that the assumption made by [Nguyen et al., 2021b] on the sufficient closeness between $p^s(y|\mathbf{z})$ and $p^t(y|\mathbf{z})$ may be stringent.

## 4.2   THE PERFORMANCE OF CS-ADV

In this section, we demonstrate how CS-adv can achieve competitive results against other SOTA methods.

**Implementation Details**    Our experiments were carried out using the PyTorch framework [Paszke et al., 2019], on an

NVIDIA GeForce RTX 3090 GPU. We use SGD optimizer with batch size 32. For Office-Home and Office-31 datasets, we resize the images to dimensions of $224 \times 224 \times 3$. We use the ResNet-50 model [He et al., 2016], pretrained on the ImageNet dataset [Deng et al., 2009], as the feature extractor, $f$. In addition, for the classifiers, $g_1$ and $g_2$, we use two fully connected layers with Leaky-ReLU activation functions (similar to [Acuna et al., 2021]). For the hyperparameters, we set $\lambda$ and $\beta$ as 1, and $\gamma$ as 0.1. For the Digit datasets, we follow the implementation in [Long et al., 2018, Acuna et al., 2021], utilizing LeNet [LeCun et al., 1998] as the backbone feature extractor, $f$. The two classifiers, $g_1$ and $g_2$, are structured identically, each comprising two linear layers, with ReLU activation functions. In our implementation, we normalize $\mathbf{z}$ and $\hat{y}$ and set kernel size $\sigma = 1$, which is a common heuristic [Greenfeld and Shalit, 2020].

**Baselines**    We compare the proposed conditional adversarial training method with some state-of-the-art domain adaptation approaches in real-world datasets that are presented in Tables 3 and 4. We compare our method with three classical adversarial training methods for domain adaptation, namely, DANN [Ganin et al., 2016] CDAN [Long et al., 2018], and MDD [Zhang et al., 2019]. We additionally compare our method with the $f$-divergence-based domain adversarial learning method, f-DAL [Acuna et al., 2021]. We consider f-DAL for comparison since it also uses a new family of divergence, $f$-divergence in an adversarial training framework. f-DAL-Alignment is a variant of f-DAL, which combines a Sampling-Based Alignment [Jiang et al., 2020] module for label shift. We further compare KL [Nguyen et al., 2021b] due to its similar motivation in offering a tighter generalization bound. We also compare our approach with Wasserstein distance-based methods (optimal transport) such as DEEPJDOT [Damodaran et al., 2018] and JUMBOT [Fatras et al., 2021], which belong to another line of related research (aligning the joint distribution $p(\mathbf{z}, y)$, but assuming $\mathbf{z}$ and $y$ are independent).

**Results**    From the results on Office-Home in Table 3, we observe that the proposed method significantly outperforms the rest of the methods in most of the adaptation tasks, and has the best performance at 71.2% on average. Also, we observe that in some tasks where the alignment for label shift leads to large improvement for f-DAL (Ar→Pr, Cl→Rw), our method yields a similar or better performance. This im-

---

[2] `https://shorturl.at/abgM0`, lines 172-175.

| Method | Ar→Cl | Ar→Pr | Ar→Rw | Cl→Ar | Cl→Pr | Cl→Rw | Pr→Ar | Pr→Cl | Pr→Rw | Rw→Ar | Rw→Cl | Rw→Pr | Avg |
|---|---|---|---|---|---|---|---|---|---|---|---|---|---|
| ResNet [He et al., 2016] | 34.9 | 50.0 | 58.0 | 37.4 | 41.9 | 46.2 | 38.5 | 31.2 | 60.4 | 53.9 | 41.2 | 59.9 | 46.1 |
| DANN [Ganin et al., 2016] | 45.6 | 59.3 | 70.1 | 47.0 | 58.5 | 60.9 | 46.1 | 43.7 | 68.5 | 63.2 | 51.8 | 76.8 | 57.6 |
| JAN [Long et al., 2017] | 45.9 | 61.2 | 68.9 | 50.4 | 59.7 | 61.0 | 45.8 | 43.4 | 70.3 | 63.9 | 52.4 | 76.8 | 58.3 |
| CDAN [Long et al., 2018] | 50.7 | 70.6 | 76.0 | 57.6 | 70.0 | 70.0 | 57.4 | 50.9 | 77.3 | 70.9 | 56.7 | 81.6 | 65.8 |
| CKB [Luo and Ren, 2021] | 54.7 | 74.4 | 77.1 | 63.7 | 72.2 | 71.8 | 64.1 | 51.7 | 78.4 | 73.1 | 58.0 | 82.4 | 68.5 |
| DEEPJDOT [Damodaran et al., 2018] | 50.7 | 68.6 | 74.4 | 59.9 | 65.8 | 68.1 | 55.2 | 46.3 | 73.8 | 66.0 | 54.9 | 78.3 | 63.5 |
| JUMBOT [Fatras et al., 2021] | 55.2 | 75.5 | 80.8 | 65.5 | 74.4 | 74.9 | 65.2 | 52.7 | 79.2 | 73.0 | 59.9 | 83.4 | 70.0 |
| MDD [Zhang et al., 2019] | 54.9 | 73.7 | 77.8 | 60.0 | 71.4 | 71.8 | 61.2 | 53.6 | 78.1 | 72.5 | 60.2 | 82.3 | 68.1 |
| f-DAL [Acuna et al., 2021] | 54.7 | 71.7 | 77.8 | 61.0 | 72.6 | 72.2 | 60.8 | 53.4 | 80.0 | **73.3** | 60.6 | **83.8** | 68.5 |
| f-DAL+Alignment [Acuna et al., 2021] | 56.7 | **77.0** | 81.1 | 63.0 | 72.2 | **75.9** | 64.5 | 54.4 | 81.0 | 72.3 | 58.4 | 83.7 | 70.0 |
| CS-adv (Ours) | **59.1** | 74.3 | **81.1** | **67.5** | **75.5** | 75.7 | **66.2** | **57.0** | **82.1** | 71.8 | **61.5** | 83.0 | **71.2** |

Table 3: Comparative results (Accuracy %) of different methods on **Office-Home**.

| Method | A→W | D→W | W→D | A→D | D→A | W→A | Avg |
|---|---|---|---|---|---|---|---|
| ResNet [He et al., 2016] | 68.4±0.2 | 96.7±0.1 | 99.3±0.1 | 68.9±0.2 | 62.5±0.3 | 60.7±0.3 | 76.1 |
| DANN [Ganin et al., 2016] | 82.0±0.4 | 96.9±0.2 | 99.1±0.1 | 79.7±0.4 | 68.2±0.4 | 67.4±0.5 | 82.2 |
| JAN [Long et al., 2017] | 85.4±0.3 | 97.4±0.2 | 99.8±0.2 | 84.7±0.3 | 68.6±0.3 | 70.0±0.4 | 84.3 |
| GTA [Sankaranarayanan et al., 2018] | 89.5±0.5 | 97.9±0.3 | 99.8±0.4 | 87.7±0.5 | 72.8±0.3 | 71.4±0.4 | 86.5 |
| MCD [Saito et al., 2018] | 88.6±0.2 | 98.5±0.1 | 100.0±.0 | 92.2±0.2 | 69.5±0.1 | 69.7±0.3 | 86.5 |
| MDD [Zhang et al., 2019] | 94.5±0.3 | 98.4±0.1 | 100.0±.0 | 93.5±0.2 | 74.6±0.3 | 72.2±0.1 | 88.9 |
| KL [Nguyen et al., 2021b] | 87.9±0.4 | 99.0±0.2 | 100.0±0.0 | 85.6±0.6 | 70.1±1.1 | 69.3±0.7 | 85.3 |
| CDAN [Long et al., 2018] | 94.1±0.1 | 98.6±0.1 | 100.0±.0 | 92.9±0.2 | 71.0±0.3 | 69.3±0.3 | 87.7 |
| f-DAL [Acuna et al., 2021] | **95.4**±0.7 | **98.8**±0.1 | **100.0**±.0 | 93.8±0.4 | 74.9±1.5 | 74.2 ±0.5 | 89.5 |
| CS-adv (Ours) | 95.1±0.6 | **98.8**±0.1 | 99.7±0.1 | **94.0**±0.5 | **76.2**±0.3 | **76.4**±0.4 | **90.0** |

Table 4: Comparative results (Accuracy %) of different methods on **Office-31**.

plies that the proposed CCS divergence has the ability to alleviate the label shift problem. The experiment results on Office-31 are shown in Table 4. It appears that the proposed method has the best performance on average. More results on Digits can be found in the Appendix (Section C.3).

### 4.3 CCS AS AN INJECTIVE MODULE

We finally provide two examples to demonstrate that the CCS divergence alone (i.e., Eq. 13) can be used as a plug-in module in existing methods.

**CCS in f-DAL**    We choose f-DAL[Acuna et al., 2021] as the first base model and simply integrate our CCS divergence into the adversarial training loss of f-DAL (f-DAL-CCS) without hyperparameter tuning. As shown in Fig. 4, CCS improves f-DAL consistently, which implies the necessity of aligning conditional distribution by CCS.

**CCS with kSHOT**    The most recent UDA approach, like kSHOT [Sun et al., 2022], uses additional prior knowledge such as the target class distribution, resulting in superior performance compared to most methods that rely solely on adversarial training. However, by integrating the CCS divergence, the performance of kSHOT is consistently improved on different tasks on Office-Home, with $0.4$ percent on average. Details can be found in Section C.4 in the Appendix.

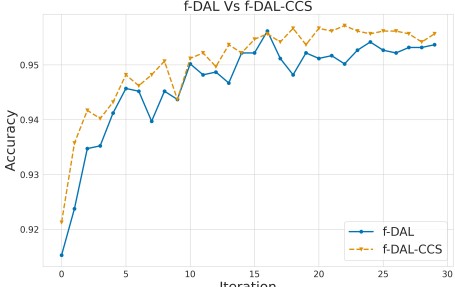

Figure 4: **Integrating CCS with f-DAL** in M→U task.

## 5   CONCLUSION

We introduce CS divergence to the problem of UDA, leading to an elegant estimation of the mismatch for both marginal (i.e., $D(p_s(\mathbf{z}); p_t(\mathbf{z}))$) and conditional distributions (i.e., $D(p_s(y|\mathbf{z}); p_t(y|\mathbf{z}))$). Compared to the MMD, it is more powerful and computationally efficient to distinguish two conditional distributions. Compared to the KL divergence, it is more stable and ensures a tighter generalization error bound. Integrating these favorable properties into a bi-classifier adversarial training framework, our method achieves SOTA performance in three UDA datasets.

Finally, our result in Eq. (9), which combines CS divergence with the fundamental Pinsker's inequality, holds the potential to tighten bounds in various other applications. Further exploration of these possibilities is left for future work.

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

# Domain Adaptation with Cauchy-Schwarz Divergence:
# Supplementary Materials

**Wenzhe Yin**[1]  **Shujian Yu**[‡2,4]  **Yicong Lin**[2]  **Jie Liu**[1]  **Jan-Jakob Sonke**[3]  **Efstratios Gavves**[1]

[1]University of Amsterdam, Amsterdam, The Netherlands
[2]Vrije Universiteit Amsterdam, Amsterdam, The Netherlands
[3]Netherlands Cancer Institute, Amsterdam, The Netherlands
[4]UiT - The Arctic University of Norway, Tromsø, Norway

## TABLE OF CONTENTS

## A  MAIN PROOFS

### A.1  PROOF OF PROPOSITION 1

**Proposition 1.** *For any $d$-variate Gaussian distributions $p \sim \mathcal{N}(\mu_1, \Sigma_1)$ and $q \sim \mathcal{N}(\mu_2, \Sigma_2)$, where $\Sigma_1$ and $\Sigma_2$ are positive definite, we have:*

$$D_{\mathrm{CS}}(p; q) \leq \min\left\{ D_{\mathrm{KL}}(p; q), D_{\mathrm{KL}}(q; p) \right\}. \tag{19}$$

*Proof.* The KL divergence for $p$ and $q$ is given by:

$$D_{\mathrm{KL}}(p; q) = \frac{1}{2}\left( \mathrm{tr}(\Sigma_2^{-1}\Sigma_1) - d + (\mu_2 - \mu_1)^\top \Sigma_2^{-1}(\mu_2 - \mu_1) + \log\left( \frac{|\Sigma_2|}{|\Sigma_1|} \right) \right), \tag{20}$$

---

[‡]Corresponding author (yusj9011@gmail.com).

where $|\cdot|$ signifies the determinant of a matrix, and $\mathrm{tr}$ denotes the trace of a matrix. Moreover, the CS divergence for $p$ and $q$ can be written as [Kampa et al., 2011]:

$$D_{\mathrm{CS}}(p;q) = -\log(z_{12}) + \frac{1}{2}\log(z_{11}) + \frac{1}{2}\log(z_{22}), \tag{21}$$

where

$$
\begin{aligned}
z_{12} &= \frac{\exp(-\frac{1}{2}(\mu_1 - \mu_2)^\top)(\Sigma_1 + \Sigma_2)^{-1}(\mu_1 - \mu_2)}{\sqrt{(2\pi)^d|\Sigma_1 + \Sigma_2|}}, \\
z_{11} &= \frac{1}{\sqrt{(2\pi)^d|2\Sigma_1|}}, \\
z_{22} &= \frac{1}{\sqrt{(2\pi)^d|2\Sigma_2|}}.
\end{aligned}
\tag{22}
$$

We can simplify the expression to:

$$
\begin{aligned}
D_{\mathrm{CS}}(p;q) &= \frac{1}{2}(\mu_2 - \mu_1)^\top(\Sigma_1 + \Sigma_2)^{-1}(\mu_2 - \mu_1) + \log\left(\sqrt{(2\pi)^d|\Sigma_1 + \Sigma_2|}\right) \\
&\quad - \frac{1}{2}\log\left(\left(\sqrt{(2\pi)^d|2\Sigma_1|}\right)\right) - \frac{1}{2}\log\left(\sqrt{(2\pi)^d|2\Sigma_2|}\right) \\
&= \frac{1}{2}(\mu_2 - \mu_1)^\top(\Sigma_1 + \Sigma_2)^{-1}(\mu_2 - \mu_1) + \frac{1}{2}\log\left(\frac{|\Sigma_1 + \Sigma_2|}{2^d\sqrt{|\Sigma_1||\Sigma_2|}}\right).
\end{aligned}
\tag{23}
$$

*Part 1.* We first consider the difference between $D_{\mathrm{CS}}(p;q)$ and $D_{\mathrm{KL}}(p;q)$ results from mean vector discrepancy, i.e., $\mu_1 - \mu_2 \neq 0_{d\times 1}$ and $\Sigma_1 = \Sigma_2 = \Sigma$. Consider two positive semi-definite Hermitian matrices $A$ and $B$ of size $n \times n$. It is known that $A - B$ is positive semi-definite if and only if $B^{-1} - A^{-1}$ is also positive semi-definite [Horn and Johnson, 2012]. Using this result, we observe that $\Sigma_2^{-1} - (\Sigma_1 + \Sigma_2)^{-1}$ is positive semi-definite, as $(\Sigma_1 + \Sigma_2) - \Sigma_2$ is positive semi-definite. Therefore, conditional on $\Sigma_1 = \Sigma_2$, we have

$$2(D_{\mathrm{CS}}(p;q) - D_{\mathrm{KL}}(p;q)) = (\mu_2 - \mu_1)^\top(\Sigma_1 + \Sigma_2)^{-1}(\mu_2 - \mu_1) - (\mu_2 - \mu_1)^\top\Sigma_2^{-1}(\mu_2 - \mu_1) \leq 0. \tag{24}$$

*Part 2.* Now we consider the difference between $D_{\mathrm{CS}}(p;q)$ and $D_{\mathrm{KL}}(p;q)$ results from covariance matrix discrepancy, i.e., $\Sigma_1 - \Sigma_2 \neq 0_{d\times d}$ and $\mu_1 = \mu_2$. We have

$$
\begin{aligned}
2(D_{\mathrm{CS}}(p;q) - D_{\mathrm{KL}}(p;q)) &= \log\left(\frac{|\Sigma_1 + \Sigma_2|}{2^d\sqrt{|\Sigma_1||\Sigma_2|}}\right) - \log\left(\frac{|\Sigma_2|}{|\Sigma_1|}\right) - \mathrm{tr}(\Sigma_2^{-1}\Sigma_1) + d. \\
&= -d\log 2 + \log\left(|\Sigma_1 + \Sigma_2|\right) - \frac{1}{2}(\log|\Sigma_1| + \log|\Sigma_2|) \\
&\quad - \log|\Sigma_2| + \log|\Sigma_1| - \mathrm{tr}(\Sigma_2^{-1}\Sigma_1) + d \\
&= -d\log 2 + \log\left(\frac{|\Sigma_1 + \Sigma_2|}{|\Sigma_2|}\right) + \frac{1}{2}\log\left(\frac{|\Sigma_1|}{|\Sigma_2|}\right) - \mathrm{tr}(\Sigma_2^{-1}\Sigma_1) + d \\
&= -d\log 2 + \log\left(|\Sigma_2^{-1}\Sigma_1 + I|\right) + \frac{1}{2}\log\left(|\Sigma_2^{-1}\Sigma_1|\right) - \mathrm{tr}(\Sigma_2^{-1}\Sigma_1) + d,
\end{aligned}
\tag{25}
$$

where $I$ represents a $d$-dimensional identity matrix. Consider the terms $|\Sigma_2^{-1}\Sigma_1|$ and $|\Sigma_2^{-1}\Sigma_1 + I|$. For convenience, let $\{\lambda_i\}_{i=1}^d$ denote the eigenvalues of $\Sigma_2^{-1}\Sigma_1$. Since $\Sigma_2^{-1}\Sigma_1$ is positive semi-definite, we have $\lambda_i \geq 0$, $i = 1\ldots, d$. We have:

$$|\Sigma_2^{-1}\Sigma_1| = \left[\left(\prod_{i=1}^d \lambda_i\right)^{1/d}\right]^d \leq \left[\frac{1}{d}\sum_{i=1}^d \lambda_i\right]^d = \left(\frac{1}{d}\mathrm{tr}(\Sigma_2^{-1}\Sigma_1)\right)^d, \tag{26}$$

where the inequality arises from the property that a geometric mean is no greater than its arithmetic counterpart. Similarly, one can have

$$|\Sigma_2^{-1}\Sigma_1 + I| = \prod_{i=1}^d (1 + \lambda_i) \leq \left[\frac{1}{d}\sum_{i=1}^d (1 + \lambda_i)\right]^d = \left(1 + \frac{1}{d}\mathrm{tr}(\Sigma_2^{-1}\Sigma_1)\right)^d. \tag{27}$$

Substituting Eqs. (26) and (27) into Eq. (25), we arrive at

$$2(D_{\mathrm{CS}}(p;q) - D_{\mathrm{KL}}(p;q)) \leq -d\log 2 + d\log\left(1 + \frac{1}{d}\mathrm{tr}(\Sigma_2^{-1}\Sigma_1)\right) + \frac{d}{2}\log\left(\frac{1}{d}\mathrm{tr}(\Sigma_2^{-1}\Sigma_1)\right) - \mathrm{tr}(\Sigma_2^{-1}\Sigma_1) + d. \quad (28)$$

We now show that $2(D_{\mathrm{CS}}(p;q) - D_{\mathrm{KL}}(p;q)) \leq 0$ conditional on $\mu_1 = \mu_2$. Let $f$ be a map given by

$$f(x) = -d\log 2 + d\log\left(1 + \frac{x}{d}\right) + \frac{d}{2}\log\left(\frac{x}{d}\right) - x + d, \qquad x \geq 0. \quad (29)$$

Since $f'(d) = 0$ and $f''(d) < 0$, we then conclude that

$$2(D_{\mathrm{CS}}(p;q) - D_{\mathrm{KL}}(p;q)) = f\left(\mathrm{tr}(\Sigma_2^{-1}\Sigma_1)\right) \leq f(d) = 0, \quad (30)$$

where $\mathrm{tr}(\Sigma_2^{-1}\Sigma_1) = \sum_{i=1}^d \lambda_i \geq 0$, conditional on $\mu_1 = \mu_2$.

*Part 3.* Note that $D_{\mathrm{CS}}(p;q) - D_{\mathrm{KL}}(p;q)$ captures the differences in both the mean vector and covariance matrix discrepancies when $\mu_1 \neq \mu_2$ and $\Sigma_1 \neq \Sigma_2$. Namely,

$$2(D_{\mathrm{CS}}(p;q) - D_{\mathrm{KL}}(p;q)) = \left[(\mu_2 - \mu_1)^\top(\Sigma_1 + \Sigma_2)^{-1}(\mu_2 - \mu_1) - (\mu_2 - \mu_1)^\top\Sigma_2^{-1}(\mu_2 - \mu_1)\right]$$
$$+ \left[\log\left(\frac{|\Sigma_1 + \Sigma_2|}{2^d\sqrt{|\Sigma_1||\Sigma_2|}}\right) - \log\left(\frac{|\Sigma_2|}{|\Sigma_1|}\right) - \mathrm{tr}(\Sigma_2^{-1}\Sigma_1) + d\right] \leq 0,$$

using Eqs. (24)) and (30).

The above analysis also applies to $D_{\mathrm{KL}}(q;p)$. That is, $2(D_{\mathrm{CS}}(p;q) - D_{\mathrm{KL}}(q;p)) \leq 0$ regardless of the parameter values. The combination of these results implies (19). $\square$

## A.2  PROOF OF PROPOSITION 2

We first present Lemma 1 that proves to be useful in Proposition 2.

**Lemma 1.** *Let $p \sim \mathcal{N}(\mu_1, \Sigma_1)$ and $q \sim \mathcal{N}(\mu_2, \Sigma_2)$ be any $d$-dimensional Gaussian distributions, the TV distance between $p$ and $q$ in case $\Sigma_1 = \Sigma_2 = \Sigma$ (positive semidefinite) can be expressed as:*

$$D_{TV} = 2\Phi\left(\frac{1}{2}\|\Sigma^{-1/2}\delta\|_2\right) - 1, \quad (31)$$

*where $\delta = \mu_1 - \mu_2$, and $\Phi$ is the cumulative distribution function of a standard normal distribution.*

*Proof.* Recall:

$$D_{\mathrm{TV}} = \frac{1}{2}\int |p(\mathbf{x}) - q(\mathbf{x})|\,\mathrm{d}\mathbf{x} = \frac{1}{2}\int\left|1 - \frac{q(\mathbf{x})}{p(\mathbf{x})}\right|p(\mathbf{x})\,\mathrm{d}\mathbf{x}. \quad (32)$$

Before continuing, we first note that for any $\mathbf{a}, \mathbf{b} \in \mathbb{R}^d$, and $S \in \mathbb{R}^{d\times d}$,

$$\mathbf{a}^\top S\mathbf{a} - \mathbf{b}^\top S\mathbf{b} = (\mathbf{a} - \mathbf{b})^\top S(\mathbf{a} - \mathbf{b}) + 2(\mathbf{a} - \mathbf{b})^\top S\mathbf{b}. \quad (33)$$

Using this identity, we have

$$\frac{q}{p}(\mathbf{x}) = \exp\left\{-\frac{1}{2}[(\mathbf{x} - \mu_2)^\top\Sigma^{-1}(\mathbf{x} - \mu_2) - (\mathbf{x} - \mu_1)^\top\Sigma^{-1}(\mathbf{x} - \mu_1)]\right\}$$
$$= \exp\left\{-\frac{1}{2}[(\mu_1 - \mu_2)^\top\Sigma^{-1}(\mu_1 - \mu_2) + 2(\mu_1 - \mu_2)^\top\Sigma^{-1}(\mathbf{x} - \mu_1)]\right\} \quad (34)$$
$$= \exp\left\{-\frac{1}{2}\|\tilde{\delta}\|_2^2 + \tilde{\delta}^\top\Sigma^{-1/2}(\mu_1 - \mathbf{x})\right\},$$

where $\tilde{\delta} = \Sigma^{-1/2}\delta$. Therefore,

$$D_{\text{TV}} = \frac{1}{2}\int \left|1 - \exp\left\{-\frac{1}{2}\|\tilde{\delta}\|_2^2 + \tilde{\delta}^\top \Sigma^{-1/2}(\mu_1 - \mathbf{x})\right\}\right| p(\mathbf{x})\,d\mathbf{x}.$$

Define a transformation $Y = \tilde{\delta}^\top \Sigma^{-1/2}(\mu_1 - X)$, where $X \sim \mathcal{N}(\mu_1, \Sigma)$. Then, $D_{\text{TV}}$ can be equivalently written as

$$D_{\text{TV}} = \frac{1}{2}\mathbb{E}_Y\left|1 - \exp\left(Y - \frac{1}{2}\|\tilde{\delta}\|_2^2\right)\right|, \tag{35}$$

where $Y \sim \mathcal{N}(0, \|\tilde{\delta}\|_2^2)$. Note that for any $Z \sim \mathcal{N}(\mu, \sigma^2)$, one can derive

$$\mathbb{E}|1 - \exp(Z)| = 1 - 2\Phi\left(\frac{\mu}{\sigma}\right) + \exp\left(\mu + \frac{1}{2}\sigma^2\right)\left(2\Phi\left(\frac{\mu+\sigma^2}{\sigma}\right) - 1\right). \tag{36}$$

Taking $\mu = -\frac{1}{2}\|\tilde{\delta}\|_2^2$ and $\sigma = \|\tilde{\delta}\|_2$ above, we have

$$D_{\text{TV}} = \frac{1}{2}\left\{1 - 2\Phi\left(-\frac{1}{2}\|\tilde{\delta}\|_2\right) + 2\Phi\left(\frac{1}{2}\|\tilde{\delta}\|_2\right) - 1\right\} = 2\Phi\left(\frac{1}{2}\|\tilde{\delta}\|_2\right) - 1 = 2\Phi\left(\frac{1}{2}\|\Sigma^{-1/2}\delta\|_2\right) - 1, \tag{37}$$

where the second equality follows from the symmetry property of $\Phi$. $\qquad\square$

Now, we proceed to the proof of Proposition 2.

**Proposition 2.** *Let $\Phi$ be the cumulative distribution function of a standard normal distribution. Let $p \sim \mathcal{N}(\mu_1, \Sigma_1)$ and $q \sim \mathcal{N}(\mu_2, \Sigma_2)$ be any $d$-dimensional Gaussian distributions. We have:*

$$D_{\text{TV}} \leq \sqrt{D_{\text{CS}}}, \tag{38}$$

*if one of the following conditions is satisfied:*

1. *$\Sigma_1 = \Sigma_2 = \Sigma$ and $1/2\sqrt{\delta^\top \Sigma^{-1}\delta} \geq 2\Phi(\|\Sigma^{-1/2}\delta\|_2/2) - 1$, where $\delta = \mu_1 - \mu_2$;*
2. *$\sum_{i=1}^d \log\left(\frac{2+\lambda_i+1/\lambda_i}{4}\right) \geq 4$, where $\lambda_i$ is the $i$-th eigenvalue of $\Sigma_2^{-1}\Sigma_1$.*

*Proof.* First, note that $D_{\text{TV}} \leq 1/2\left(\int p(\mathbf{x})\,d\mathbf{x} + \int q(\mathbf{x})\,d\mathbf{x}\right) \leq 1$, whereas $D_{\text{CS}}$ is unbounded and can easily exceed values of 1. Recall $\delta = \mu_1 - \mu_2$. The closed-form expression of CS divergence is:

$$D_{\text{CS}}(p; q) = \frac{1}{2}\delta^\top(\Sigma_1 + \Sigma_2)^{-1}\delta + \frac{1}{2}\log\left(\frac{|\Sigma_1 + \Sigma_2|}{2^d\sqrt{|\Sigma_1||\Sigma_2|}}\right), \tag{39}$$

where the first term and the second term quantify the discrepancy resulting from the difference of mean vectors and covariance matrices, respectively.

*Part 1.* Consider $\Sigma_1 = \Sigma_2 = \Sigma$. By Lemma 1, we have $D_{\text{TV}} = 2\Phi(\|\Sigma^{-1/2}\delta\|_2/2) - 1$. Hence,

$$D_{\text{TV}} \leq \sqrt{D_{\text{CS}}} \iff \frac{1}{2}\sqrt{\delta^\top \Sigma_1^{-1}\delta} \geq 2\Phi(\|\Sigma_1^{-1/2}\delta\|_2/2) - 1. \tag{40}$$

*Part 2.* More generally, given that the TV distance for two Gaussian distributions lacks a closed-form expression, it suffices to examine the conditions under which $D_{\text{CS}} \geq 1$. We have

$$\begin{aligned}
D_{\text{CS}} &\geq \frac{1}{2}\ln\left(\frac{|\Sigma_1 + \Sigma_2|}{2^d\sqrt{|\Sigma_1||\Sigma_2|}}\right) \\
&= \frac{1}{2}\left(\frac{1}{2}\ln\left(\frac{|\Sigma_1 + \Sigma_2|}{|\Sigma_1|}\right) + \frac{1}{2}\ln\left(\frac{|\Sigma_1 + \Sigma_2|}{|\Sigma_2|}\right) - d\ln 2\right) \\
&= \frac{1}{2}\left(\frac{1}{2}\ln\left(|\Sigma_1^{-1}\Sigma_2 + I|\right) + \frac{1}{2}\ln\left(|\Sigma_2^{-1}\Sigma_1 + I|\right) - d\ln 2\right).
\end{aligned} \tag{41}$$

Let $\lambda_i$ denotes the $i$-th eigenvalue of $\Sigma_2^{-1}\Sigma_1$, then $1/\lambda_i$ is the $i$-th eigenvalue of $\Sigma_1^{-1}\Sigma_2$. We have

$$|\Sigma_1^{-1}\Sigma_2 + I| = \prod_{i=1}^{d}(1/\lambda_i + 1), \qquad |\Sigma_2^{-1}\Sigma_1 + I| = \prod_{i=1}^{d}(\lambda_i + 1). \tag{42}$$

It leads to

$$D_{\text{CS}} \geq \frac{1}{2}\sum_{i=1}^{d}\left(\frac{1}{2}\ln(\lambda_i + 1) + \frac{1}{2}\ln(1/\lambda_i + 1) - \ln 2\right) = \frac{1}{4}\sum_{i=1}^{d}\log\left(\frac{2 + \lambda_i + 1/\lambda_i}{4}\right). \tag{43}$$

Given the condition $\sum_{i=1}^{d}\log\left(\frac{2+\lambda_i+1/\lambda_i}{4}\right) \geq 4$, we have $D_{\text{TV}} \leq 1 \leq \sqrt{D_{\text{CS}}}$. The proof is now completed. $\qquad\square$

**Remark 3.** *In fact, the conditions outlined in Proposition 2 are easily satisfied, particularly when $p$ and $q$ exhibit significant dissimilarity, and the variable dimension $d$ is large. For simplicity, let's consider a diagonal covariance matrix $\Sigma = \text{diag}\left(\sigma_1^2, \sigma_2^2, \ldots, \sigma_d^2\right)$. In this case, the condition $1/2\sqrt{\delta^{\top}\Sigma_1^{-1}\delta} \geq 2\Phi(\|\Sigma_1^{-1/2}\delta\|_2/2) - 1$ reduces to:*

$$\frac{1}{2}\sqrt{\sum_{i=1}^{d}\left(\frac{\delta_i}{\sigma_i}\right)^2} \geq 2\Phi\left(\frac{\sqrt{\sum_{i=1}^{d}(\delta_i/\sigma_i)^2}}{2}\right) - 1. \tag{44}$$

*The R.H.S. of Eq. (44) is upper bounded by 1, whereas the L.H.S. of Eq. (44) is unbounded and is prone to increase with the addition of new dimension (if other dimensions remain unchanged). On the other hand, since $\log\left(\frac{2+\lambda_i+1/\lambda_i}{4}\right) \geq 0$, the L.H.S. of Eq. (43) is unbounded and is prone to increase with the addition of new dimension (if $\lambda_i$, $i = 1, 2, .., d-1$, remain unchanged).*

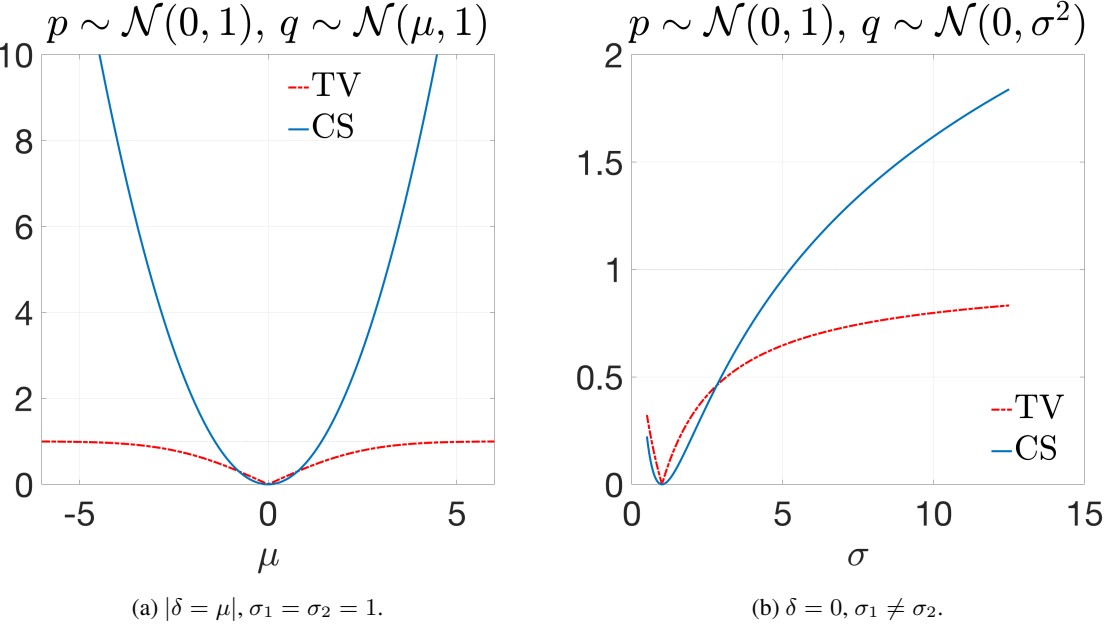

(a) $|\delta = \mu|$, $\sigma_1 = \sigma_2 = 1$.          (b) $\delta = 0$, $\sigma_1 \neq \sigma_2$.

Figure 5: Values of $D_{\text{TV}}$ and $\sqrt{D_{\text{CS}}}$ for 1-dimensional Gaussian data in case (a) $\mu$ is different, $\sigma > 0$ is the same; and (b) $\sigma$ is different, $\mu$ is the same.

*Moreover, from Fig. 5, it is easy to observe that, when $d = 1$, the TV distance is too conservative and quickly reaches its upper bound 1, whereas the CS divergence is unbounded and larger than the TV distance if $p$ and $q$ are not sufficiently similar.*

## A.3 PROOF OF PROPOSITION 3

We first present a lemma (without proof), referred to as the Jensen weighted integral inequality, which proves to be useful in the subsequent proof.

**Lemma 2.** *[Dragomir et al., 2003] Assume a convex function $f : I \mapsto \mathbb{R}$. Moreover, $g, h : [x_1, x_2] \mapsto \mathbb{R}$ are measurable functions such that $g(x) \in I$ and $h(x) \geq 0, \forall x \in [x_1, x_2]$. Also suppose that $h$, $gh$, and $(f \circ g) \cdot h$ are all integrable functions on $[x_1, x_2]$ and $\int_{x_1}^{x_2} h(x)\, \mathrm{d}x > 0$, then*

$$f\left( \frac{\int_{x_1}^{x_2} g(x)h(x)\, \mathrm{d}x}{\int_{x_1}^{x_2} h(x)\, \mathrm{d}x} \right) \leq \frac{\int_{x_1}^{x_2} (f \circ g)(x)h(x)\, \mathrm{d}x}{\int_{x_1}^{x_2} h(x)\, \mathrm{d}x}. \tag{45}$$

Let $f(x) = x \log(x)$, which is a convex function. For some positive functions $a, b$, set $h = b$ and $g = a/b$ in Lemma 2. We have

$$\left( \int_{x_1}^{x_2} a(x)\, \mathrm{d}x \right) \log \left( \frac{\int_{x_1}^{x_2} a(x)\, \mathrm{d}x}{\int_{x_1}^{x_2} b(x)\, \mathrm{d}x} \right) \leq \int_{x_1}^{x_2} a(x) \log \frac{a(x)}{b(x)}\, \mathrm{d}x. \tag{46}$$

The inequality above holds for any integration range, provided the Riemann integrals exist. Moreover, this inequality can be easily extended to general ranges, including possibly disconnected sets, using Lebesgue integrals. In fact, Eq. (46) can be understood as a continuous extension of the well-known log sum inequality. For simplicity, we denote $\int_{x_1}^{x_2} a(x)\, \mathrm{d}x = \int_K a(x)\, \mathrm{d}x$, where $|K| = x_2 - x_1 \gg 0$ refers to the length of the integral's interval.

**Proposition 3.** *For any density functions $p : \mathbb{R}^d \to \mathbb{R}_{\geq 0}$ and $q : \mathbb{R}^d \to \mathbb{R}_{\geq 0}$, let $K$ be an integration domain over which $p$ and $q$ are Riemann integrable. Suppose $|K| < \infty$, where $|K|$ denotes the volume. Then*

$$C_1 \left[ D_{\mathrm{CS}}(p; q) - \log |K| + 2 \log C_2 \right] \leq D_{\mathrm{KL}}(p; q), \tag{47}$$

*where $C_1 = \int_K p(\mathbf{x})\, \mathrm{d}\mathbf{x}$, $C_2 = C_1 \left( \int_K p^2(\mathbf{x})\, \mathrm{d}\mathbf{x} \int_K q^2(\mathbf{x})\, \mathrm{d}\mathbf{x} \right)^{-1/4}$. Clearly, for $K$ such that $|K \cap S| \gg 0$, where $S = \{ \mathbf{x} : p(\mathbf{x}) > 0 \}$, one can have $C_1 \approx 1$.*

*Proof.* The following results hold for multivariate density functions. Without loss of generality, we focus on the univariate case. Construct the following two functions:

$$a(x) = p(x)/C_2, \qquad b(x) = \sqrt{p(x)q(x)}. \tag{48}$$

Clearly, $\sqrt{p(x)/q(x)} = \left( a(x)/b(x) \right) C_2$. We have

$$
\begin{aligned}
D_{\mathrm{KL}}(p; q) &= \int_K p(x) \log \frac{p(x)}{q(x)}\, \mathrm{d}x \\
&= 2 \int_K p(x) \log \sqrt{\frac{p(x)}{q(x)}}\, \mathrm{d}x \\
&= 2 \int_K a(x) C_2 \log \left( \frac{a(x)}{b(x)} C_2 \right)\, \mathrm{d}x \\
&= 2C_2 \left[ \int_K a(x) \log \left( \frac{a(x)}{b(x)} \right)\, \mathrm{d}x + \log C_2 \int_K a(x)\, \mathrm{d}x \right] \\
&\geq 2C_2 \left[ \left( \int_K a(x)\, \mathrm{d}x \right) \log \left( \frac{\int_K a(x)\, \mathrm{d}x}{\int_K b(x)\, \mathrm{d}x} \right) + \log C_2 \int_K a(x)\, \mathrm{d}x \right] \\
&= 2C_2 \int_K a(x)\, \mathrm{d}x \left[ \log \left( \frac{\int_K a(x)\, \mathrm{d}x}{\int_K b(x)\, \mathrm{d}x} \right) + \log C_2 \right],
\end{aligned}
\tag{49}
$$

where the inequality is due to Eq. (46). Note that

$$\int_K a(x)\, \mathrm{d}x = \int_K \frac{p(x)}{C_2}\, \mathrm{d}x = \frac{1}{C_2} \int_K p(x)\, \mathrm{d}x = \left( \int_K p^2(x)\, \mathrm{d}x \int_K q^2(x)\, \mathrm{d}x \right)^{1/4}, \tag{50}$$

and, using the Cauchy-Schwarz inequality,

$$\left(\int_K b(x)\,\mathrm{d}x\right)^2 = \left(\int_K \sqrt{p(x)q(x)} \cdot 1\,\mathrm{d}x\right)^2 \leq \left(\int_K p(x)q(x)\,\mathrm{d}x\right)\left(\int_K 1\,\mathrm{d}x\right) = \left(\int_K p(x)q(x)\,\mathrm{d}x\right)|K|. \quad (51)$$

Substituting (50) and (51) into (49), we have

$$
\begin{aligned}
D_{\mathrm{KL}}(p;q) &\geq 2C_2 \int_K a(x)\,\mathrm{d}x \left[\log\left(\frac{\int_K a(x)\,\mathrm{d}x}{\int_K b(x)\,\mathrm{d}x}\right) + \log C_2\right] \\
&= C_1\left[\log\left(\frac{\int_K a(x)\,\mathrm{d}x}{\int_K b(x)\,\mathrm{d}x}\right)^2 + 2\log C_2\right] \\
&= C_1\left[\log\left(\frac{\left(\int_K p^2(x)\,\mathrm{d}x \int_K q^2(x)\,\mathrm{d}x\right)^{1/2}}{\left(\int_K b(x)\,\mathrm{d}x\right)^2}\right) + 2\log C_2\right] \\
&\geq C_1\left[\log\left(\frac{\left(\int_K p^2(x)\,\mathrm{d}x \int_K q^2(x)\,\mathrm{d}x\right)^{1/2}}{\left(\int_K p(x)q(x)\right)|K|}\right) + 2\log C_2\right] \\
&= C_1\left[D_{\mathrm{CS}}(p;q) - \log|K| + 2\log C_2\right].
\end{aligned}
\quad (52)
$$

The proof is completed. $\qquad\square$

## A.4  PROOF OF PROPOSITION 4

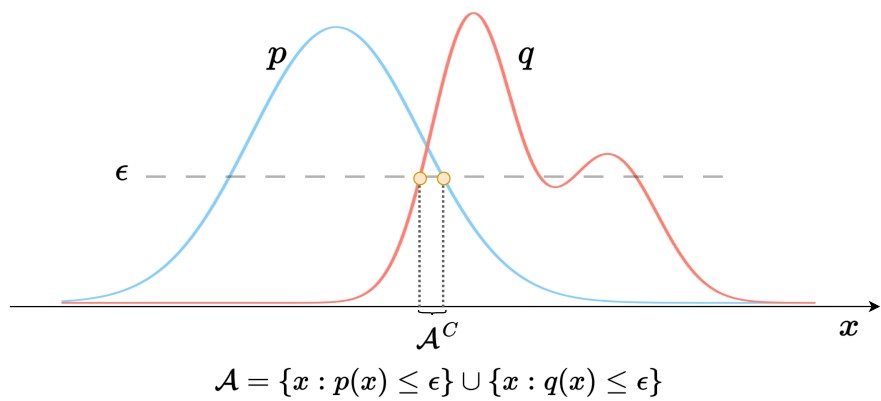

$$\mathcal{A} = \{x : p(x) \leq \epsilon\} \cup \{x : q(x) \leq \epsilon\}$$

Figure 6: A graphical illustration of the sets $\mathcal{A}_\epsilon$ and $\mathcal{A}_\epsilon^{\mathsf{C}}$ defined in Proposition 4.

**Proposition 4.** *For any density functions $p$ and $q$, and any $\epsilon > 0$, let $\mathcal{A}_\epsilon = \{\mathbf{x} : p(\mathbf{x}) \leq \epsilon\} \cup \{\mathbf{x} : q(\mathbf{x}) \leq \epsilon\}$ and $\mathcal{A}_\epsilon^{\mathsf{C}}$ be its complement. Moreover, define $T_{\mathcal{A}_\epsilon^{\mathsf{C}}} = \sup\left\{p(\mathbf{x})q(\mathbf{x}),\, \mathbf{x} \in \mathcal{A}_\epsilon^{\mathsf{C}}\right\}$ and $\left|\mathcal{A}_\epsilon^{\mathsf{C}}\right|$ to denote the "length" of the set $\mathcal{A}_\epsilon^{\mathsf{C}}$ (strictly speaking, the Lebesgue measure of the set $\mathcal{A}_\epsilon^{\mathsf{C}}$). Suppose there exists an $\epsilon > 0$ such that $T_{\mathcal{A}_\epsilon^{\mathsf{C}}}\left|\mathcal{A}_\epsilon^{\mathsf{C}}\right| < \infty$ and $C_3 = \int p^2(\mathbf{x})\,\mathrm{d}\mathbf{x} \int q^2(\mathbf{x})\,\mathrm{d}\mathbf{x} \geq \exp(2)\left(2\epsilon + T_{\mathcal{A}_\epsilon^{\mathsf{C}}}\left|\mathcal{A}_\epsilon^{\mathsf{C}}\right|\right)^2$, then*

$$D_{\mathrm{TV}}(p;q) \leq \sqrt{D_{\mathrm{CS}}(p;q)}. \quad (53)$$

*Proof.* Note that $D_{\text{CS}}(p;q) = -\log(\int p(\mathbf{x})q(\mathbf{x})\,d\mathbf{x}) + 1/2\log(C_3)$. For the term $\int p(\mathbf{x})q(\mathbf{x})\,d\mathbf{x}$, we can write

$$
\begin{aligned}
\int p(\mathbf{x})q(\mathbf{x})\,d\mathbf{x} &= \int_{\mathcal{A}_\epsilon} p(\mathbf{x})q(\mathbf{x})\,d\mathbf{x} + \int_{\mathcal{A}_\epsilon^{\complement}} p(\mathbf{x})q(\mathbf{x})\,d\mathbf{x} \\
&\leq \int_{\mathcal{A}_\epsilon} \epsilon \max\{p(\mathbf{x}),q(\mathbf{x})\}\,d\mathbf{x} + \int_{\mathcal{A}_\epsilon^{\complement}} p(\mathbf{x})q(\mathbf{x})\,d\mathbf{x} \\
&\leq \epsilon \int (p(\mathbf{x})+q(\mathbf{x}))\,d\mathbf{x} + \int_{\mathcal{A}_\epsilon^{\complement}} p(\mathbf{x})q(\mathbf{x})\,d\mathbf{x} \\
&= 2\epsilon + \int_{\mathcal{A}_\epsilon^{\complement}} p(\mathbf{x})q(\mathbf{x})\,d\mathbf{x} \\
&\leq 2\epsilon + T_{\mathcal{A}_\epsilon^{\complement}} \left| \mathcal{A}_\epsilon^{\complement} \right|.
\end{aligned}
\tag{54}
$$

Hence, $D_{\text{CS}}(p;q) \geq -\log\left(2\epsilon + T_{\mathcal{A}_\epsilon^{\complement}} \left| \mathcal{A}_\epsilon^{\complement} \right|\right) + 1/2\log(C_3) \geq 1 \geq D_{\text{TV}}(p;q)$. $\qquad\square$

**Remark 4.** *We provide some explanation of the conditions in Proposition 4. These conditions imply that as two densities $p, q$ exhibit less and less overlap (i.e., in the case of a small $\epsilon > 0$) the integral of $pq$ tends toward 0. Consequently, $-\log\left(\int p(x)q(x)dx\right) \gg 0$ in $D_{\text{CS}}(p;q)$ dominates $\log\left(\int p^2(x)dx\right) + \log\left(\int q^2(x)dx\right)$ because $\int p^2(x)dx$ and $\int q^2(x)dx$ are constants unaffected by the extent of overlap between $p$ and $q$. Therefore, $D_{\text{CS}}(p;q)$ can rapidly surpass 1 when the shapes of $p$ and $q$ are markedly distinct.*

*For illustration, let $p$ be the pdf of $\mathcal{N}(\mu_1,\sigma_1^2)$ and $q$ be the pdf of $\mathcal{N}(\mu_2,\sigma_2^2)$. For $\epsilon > 0$, we consider two examples: (i) $\mu_2 = \mu_1 + \delta_\epsilon > \mu_1$ and $\sigma_1 = \sigma_2 = \sigma > 0$; (ii) $\mu_1 = \mu_2 = \mu$ and $\sigma_2 = \sigma_1 + \delta_\epsilon > \sigma_1$, where $\delta_\epsilon > 0$ relies on $\epsilon$.*

*(i) For $\epsilon > 0$, we have*

$$
\begin{aligned}
\mathcal{A} &= \left(-\infty, \mu_2 - \sigma\sqrt{-\log(2\pi\sigma^2\epsilon^2)}\right] \cup \left[\mu_1 + \sigma\sqrt{-\log(2\pi\sigma^2\epsilon^2)}, +\infty\right), \\
\left|\mathcal{A}^{\complement}\right| &= 2\sigma\sqrt{-\log(2\pi\sigma^2\epsilon^2)} - \delta_\epsilon, \\
T_{\mathcal{A}^{\complement}} &\leq \left(2\pi\sigma^2\right)^{-1}\exp\left(-\frac{(2\mu_1+\delta_\epsilon)^2}{4\sigma^2}\right), \\
C_3 &= \left(4\pi\sigma^2\right)^{-1}.
\end{aligned}
$$

*It is not hard to see that for any $\epsilon > 0$, when $\delta_\epsilon$ is sufficiently large, indicating that $p$ and $q$ substantially differ from each other, one can achieve $T_{\mathcal{A}^{\complement}}\left|\mathcal{A}^{\complement}\right| \leq 2\epsilon$ because $T_{\mathcal{A}^{\complement}}$ decays to 0 exponentially fast when $\delta_\epsilon$ increases. Additionally, satisfying $C_3 \geq \exp(2)\left(2\epsilon + T_{\mathcal{A}_\epsilon^{\complement}}\left|\mathcal{A}_\epsilon^{\complement}\right|\right)^2$ is not challenging if $\epsilon$ is chosen small.*

*(ii) Similarly, for $\epsilon > 0$, we have*

$$
\begin{aligned}
\mathcal{A} &= \left(-\infty, \mu_1 - \sigma_1\sqrt{-\log(2\pi\sigma_1^2\epsilon^2)}\right] \cup \left[\mu_1 + \sigma_1\sqrt{-\log(2\pi\sigma_1^2\epsilon^2)}, +\infty\right), \\
\left|\mathcal{A}^{\complement}\right| &= 2\sigma_1\sqrt{-\log(2\pi\sigma_1^2\epsilon^2)}, \\
T_{\mathcal{A}^{\complement}} &\leq \left[2\pi\sigma_1(\sigma_1+\delta_\epsilon)\right]^{-1}, \\
C_3 &= \left[4\pi\sigma_1(\sigma_1+\delta_\epsilon)\right]^{-1}.
\end{aligned}
$$

*As before, for any $\epsilon > 0$, as long as $\delta_\epsilon$ is sufficiently large, we can have $T_{\mathcal{A}^{\complement}}\left|\mathcal{A}^{\complement}\right| \leq 2\epsilon$ and $C_3 \geq \exp(2)\left(2\epsilon + T_{\mathcal{A}_\epsilon^{\complement}}\left|\mathcal{A}_\epsilon^{\complement}\right|\right)^2$.*

## A.5 EMPIRICAL VALIDATION OF RELATIONSHIP BETWEEN DIFFERENT DIVERGENCE MEASURES

We finally provide an empirical validation to show that the following relationship generally holds:

$$
D_{\text{TV}} \lesssim \sqrt{D_{\text{CS}}} \quad \text{and} \quad D_{\text{CS}} \lesssim D_{\text{KL}},
\tag{55}
$$

where $p$ and $q$ need not be Gaussian, and the symbol $\lesssim$ denotes "less than or similar to".

We start our analysis for discrete $p$ and $q$ for simplicity. This is because, unlike the CS divergence, TV distance and KL divergence do not have closed-form expressions for neither Gaussian distributions nor a mixture-of-Gaussian (MoG) [Devroye et al., 2018]. Hence, it becomes challenging to perform Monte Carlo simulations for continuous cases.

Consider the probability mass functions $p$ and $q$ with the support $\mathcal{X} = \{x_1, x_2, \ldots, x_K\}$ (i.e., there are $K$ different discrete states). Namely, $\sum_{i=1}^{K} p(x_i) = \sum_{i=1}^{K} q(x_i) = 1$. We have

$$D_{\text{TV}}(p; q) = \frac{1}{2} \sum |p(x_i) - q(x_i)|, \tag{56}$$

$$D_{\text{CS}}(p; q) = -\log \left( \frac{\sum p(x_i) q(x_i)}{\sqrt{\sum p(x_i)^2} \sqrt{\sum q(x_i)^2}} \right), \tag{57}$$

$$D_{\text{KL}}(p; q) = \sum_{i=1}^{K} p(x_i) \log \left( \frac{p(x_i)}{q(x_i)} \right). \tag{58}$$

For some $K$, we randomly generate probability pairs $\{(p_i, q_i),\ i = 1, \ldots K\}$. Fig. 7 demonstrates the values of $D_{\text{TV}}$ with respect to $\sqrt{D_{\text{CS}}}$ (first row) and $D_{\text{KL}}$ with respect to $D_{\text{CS}}$ (second row) when $K = 2$ (first column) $K = 3$ (second column) and $K = 10$ (third column), respectively. We only show results of $1,000$ replicates.

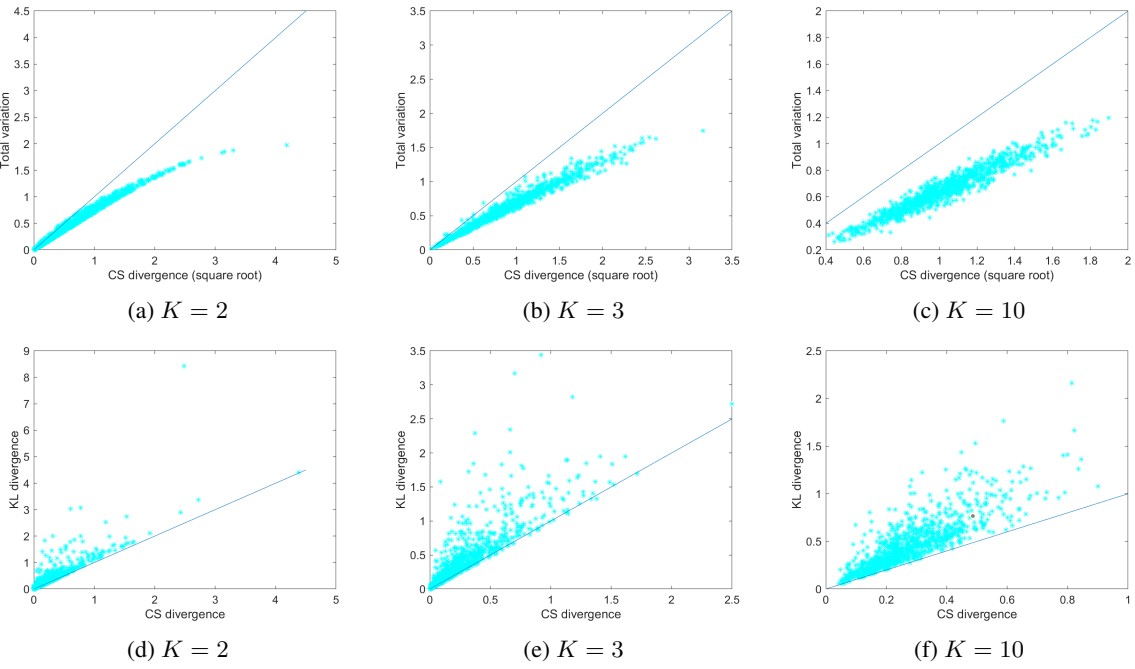

Figure 7: Values of $D_{\text{TV}}$ with respect to $\sqrt{D_{\text{CS}}}$ (first row) and $D_{\text{KL}}$ with respect to $D_{\text{CS}}$ (second row) for $1,000$ replicates of randomly generated probability pairs $\{(p_i, q_i),\ i = 1, \ldots K\}$, when $K = 2$ (first column), $K = 3$ (second column), and $K = 10$ (third column). The diagonal indicates $D_{\text{TV}} = \sqrt{D_{\text{CS}}}$ or $D_{\text{KL}} = D_{\text{CS}}$.

## B  EMPIRICAL ESTIMATOR OF CS AND CONDITIONAL CS

We now use subscripts to denote the domain index for notational convenience.

## B.1 EMPIRICAL ESTIMATOR OF CS

**Proposition 5** (Empirical Estimator of $D_{\text{CS}}(p_s(\mathbf{z}); p_t(\mathbf{z}))$ [Jenssen et al., 2006]). *Given observations $\{\mathbf{z}_i^s\}_{i=1}^M$ and $\{\mathbf{z}_i^t\}_{i=1}^N$, the empirical estimator of $D_{\text{CS}}(p_s(\mathbf{z}); p_t(\mathbf{z}))$ is given by:*

$$\widehat{D}_{\text{CS}}(p_s(\mathbf{z}); p_t(\mathbf{z})) = \log \frac{1}{M^2} \sum_{i,j=1}^M \kappa(\mathbf{z}_i^s, \mathbf{z}_j^s)) +$$

$$\log(\frac{1}{N^2} \sum_{i,j=1}^N \kappa(\mathbf{z}_i^t, \mathbf{z}_j^t)) - 2\log(\frac{1}{MN} \sum_{i=1}^M \sum_{j=1}^N \kappa(\mathbf{z}_i^s, \mathbf{z}_j^t)). \tag{59}$$

*where $\kappa$ is a kernel function such as Gaussian $\kappa_\sigma(\mathbf{z}, \mathbf{z}') = \exp(-\|\mathbf{z} - \mathbf{z}'\|_2^2/2\sigma^2)$.*

*Proof.* The CS divergence is defined by:

$$D_{\text{CS}}(p_s; p_t) = -\log\left(\frac{(\int p_s(\mathbf{z})p_t(\mathbf{z})\,\mathrm{d}\mathbf{z})^2}{\int p_s(\mathbf{z})^2\,\mathrm{d}\mathbf{z} \int p_t(\mathbf{z})^2\,\mathrm{d}\mathbf{z}}\right). \tag{60}$$

By the kernel density estimation (KDE), we have:

$$\hat{p}_s(\mathbf{z}) = \frac{1}{M} \sum_{i=1}^M \kappa_\sigma(\mathbf{z} - \mathbf{z}_i^s), \tag{61}$$

and

$$\hat{p}_t(\mathbf{z}) = \frac{1}{N} \sum_{i=1}^N \kappa_\sigma(\mathbf{z} - \mathbf{z}_i^t). \tag{62}$$

By substituting Eq. (61) into $\int \hat{p}_s^2(\mathbf{z})\,\mathrm{d}z$, we have:

$$\int \hat{p}_s^2(\mathbf{z})\,\mathrm{d}z = \int \left(\frac{1}{M} \sum_{i=1}^M \kappa_\sigma(\mathbf{z} - \mathbf{z}_i^s)\right)^2 \,\mathrm{d}z$$

$$= \frac{1}{M^2} \int \left(\sum_{i=1}^M \sum_{j=1}^M \kappa_\sigma(\mathbf{z} - \mathbf{z}_j^s) \cdot \kappa_\sigma(\mathbf{z} - \mathbf{z}_i^s)\right) \,\mathrm{d}z$$

$$= \frac{1}{M^2} \sum_{i=1}^M \sum_{j=1}^M \int \kappa_\sigma(\mathbf{z} - \mathbf{z}_j^s) \cdot \kappa_\sigma(\mathbf{z} - \mathbf{z}_i^s) \,\mathrm{d}z$$

$$= \frac{1}{M^2} \sum_{i=1}^M \sum_{j=1}^M \kappa_{\sqrt{2}\sigma}(\mathbf{z}_j^s - \mathbf{z}_i^s). \tag{63}$$

The final equation is derived by using the property that the integral of the product of two Gaussians equals the value of the Gaussian computed at the difference of the arguments with the variance being the sum of the variances of the two original Gaussian functions [Bromiley, 2003].

Similarly,

$$\int \hat{p}_t^2(\mathbf{z})\,\mathrm{d}z = \frac{1}{N^2} \sum_{i=1}^N \sum_{j=1}^N \kappa_{\sqrt{2}\sigma}(\mathbf{z}_j^t - \mathbf{z}_i^t), \tag{64}$$

and

$$\int \hat{p}_s(\mathbf{z})\hat{p}_t(\mathbf{z})\,\mathrm{d}z = \frac{1}{MN} \sum_{i=1}^M \sum_{j=1}^N \kappa_{\sqrt{2}\sigma}(\mathbf{z}_j^t - \mathbf{z}_i^s). \tag{65}$$

By substituting Eqs. (63)-(65) into the definition of CS divergence in Eq. (60), we obtain:

$$
\widehat{D}_{\text{CS}}(p_a; p_t) = \log \left( \frac{1}{M^2} \sum_{i,j=1}^{M} \kappa_{\sqrt{2}\sigma}(\mathbf{z}_j^s - \mathbf{z}_i^s) \right) + \log \left( \frac{1}{N^2} \sum_{i,j=1}^{N} \kappa_{\sqrt{2}\sigma}(\mathbf{z}_j^t - \mathbf{z}_i^t) \right)
$$
$$
- 2 \log \left( \frac{1}{MN} \sum_{i=1}^{M} \sum_{j=1}^{N} \kappa_{\sqrt{2}\sigma}(\mathbf{z}_j^t - \mathbf{z}_i^s) \right). \tag{66}
$$

**Connection to MMD** Interestingly, we found the CS divergence is closely related to the MMD. Here, we demonstrate the connection between the CS divergence and MMD. A natural choice for measuring the dissimilarity between $p_s$ and $p_t$ is the Euclidean distance:

$$
D_{\text{ED}}(p_s; p_t) = \int (\hat{p}_s(\mathbf{z}) - \hat{p}_t(\mathbf{z}))^2 \, \mathrm{d}z
$$
$$
= \int \hat{p}_s^2(\mathbf{z}) \, \mathrm{d}z + \int \hat{p}_t^2(\mathbf{z}) \, \mathrm{d}z - \int \hat{p}_s(\mathbf{z})\hat{p}_t(\mathbf{z}) \, \mathrm{d}z \tag{67}
$$

Combining Eqs. (63)-(65), we have:

$$
D_{\text{ED}}(p_s; p_t) = \frac{1}{M^2} \sum_{i,j=1}^{M} \kappa_{\sqrt{2}\sigma}(\mathbf{z}_j^s - \mathbf{z}_i^s) + \frac{1}{N^2} \sum_{i,j=1}^{N} \kappa_{\sqrt{2}\sigma}(\mathbf{z}_j^t - \mathbf{z}_i^t)
$$
$$
- \frac{2}{MN} \sum_{i,j=1}^{M,N} \kappa_{\sqrt{2}\sigma}(\mathbf{z}_j^t - \mathbf{z}_i^s). \tag{68}
$$

Note that Eq. (68) is exactly the same (in terms of mathematical expression) as the square of MMD using $V$-statistic estimator [Gretton et al., 2012]:

$$
\widehat{\text{MMD}}_v[p_s(\mathbf{z}), p_t(\mathbf{z})] = \left[ \frac{1}{M^2} \sum_{i,j=1}^{M} \kappa(\mathbf{z}_i^s, \mathbf{z}_j^s) + \frac{1}{N^2} \sum_{i,j=1}^{N} \kappa(\mathbf{z}_i^t, \mathbf{z}_j^t) - \frac{2}{MN} \sum_{i,j=1}^{M,N} \kappa(\mathbf{z}_j^t, \mathbf{z}_i^s) \right]^{\frac{1}{2}}, \tag{69}
$$

by using a Gaussian kernel $\kappa$ with variance $\sqrt{2}\sigma$. Also, we have:

$$
\widehat{\text{MMD}}^2(p^s; p^t) = \langle \mu_s, \mu_t \rangle_{\mathcal{H}}^2 = \frac{1}{M^2} \sum_{i,j=1}^{M} \kappa(\mathbf{z}_i^s, \mathbf{z}_j^s) + \frac{1}{N^2} \sum_{i,j=1}^{N} \kappa(\mathbf{z}_i^t, \mathbf{z}_j^t) - \frac{2}{MN} \sum_{i=1}^{M} \sum_{j=1}^{N} \kappa(\mathbf{z}_i^s, \mathbf{x}_j^t) \tag{70}
$$

By comparing Eq. (59) with Eq. (70), it is interesting to find that the empirical estimator of CS divergence just adds a logarithm on each term of that of MMD.

$\square$

## B.2  EMPIRICAL ESTIMATOR OF CCS

The conditional CS divergence for $p_s(\mathbf{y}|\mathbf{z})$ and $p_t(\mathbf{y}|\mathbf{z})$ is expressed as:

$$
D_{\text{CS}}(p_s(\mathbf{y}|\mathbf{z}); p_t(\mathbf{y}|\mathbf{z})) =
$$
$$
- 2 \log \left( \int_{\mathcal{Z}} \int_{\mathcal{Y}} \frac{p_s(\mathbf{z}, \mathbf{y}) p_t(\mathbf{z}, \mathbf{y})}{p_s(\mathbf{z}) p_t(\mathbf{z})} \, \mathrm{d}\mathbf{z}\mathrm{d}\mathbf{y} \right) + \log \left( \int_{\mathcal{Z}} \int_{\mathcal{Y}} \frac{p_s^2(\mathbf{z}, \mathbf{y})}{p_s^2(\mathbf{z})} \, \mathrm{d}\mathbf{z}\mathrm{d}\mathbf{y} \right)
$$
$$
+ \log \left( \int_{\mathcal{Z}} \int_{\mathcal{Y}} \frac{p_t^2(\mathbf{z}, \mathbf{y})}{p_t^2(\mathbf{z})} \, \mathrm{d}\mathbf{z}\mathrm{d}\mathbf{y} \right). \tag{71}
$$

which contains two conditional quadratic terms (i.e., $\int_{\mathcal{Z}} \int_{\mathcal{Y}} \frac{p_s^2(\mathbf{z},\mathbf{y})}{p_s^2(\mathbf{z})} \, \mathrm{d}\mathbf{z}\mathrm{d}\mathbf{y}$ and $\int_{\mathcal{Z}} \int_{\mathcal{Y}} \frac{p_t^2(\mathbf{z},\mathbf{y})}{p_t^2(\mathbf{z})} \, \mathrm{d}\mathbf{z}\mathrm{d}\mathbf{y}$) and a cross term (i.e., $\int_{\mathcal{Z}} \int_{\mathcal{Y}} \frac{p_s(\mathbf{z},\mathbf{y})p_t(\mathbf{z},\mathbf{y})}{p_s(\mathbf{z})p_t(\mathbf{z})} \, \mathrm{d}\mathbf{z}\mathrm{d}\mathbf{y}$). Note, we use $y$ instead of $\hat{y}$ in Proposition 2 in the main manuscript to represent label for the convenience and clear demonstration.

**Proposition 6** (Empirical Estimator of $D_{\text{CCS}}(p_s(y|\mathbf{z}); p_t(y|\mathbf{z}))$). *Given observations $\{\mathbf{z}_i^s, y_i^s\}_{i=1}^M$ and $\{\mathbf{z}_i^t, y_i^t\}_{i=1}^N$. Let $K^s$ and $L^s$ denote, respectively, the Gram matrices for the variable $\mathbf{z}$ and the predicted output $\hat{y}$ in the source distribution. Similarly, let $K^t$ and $L^t$ denote, respectively, the Gram matrices for the variable $\mathbf{z}$ and the label $y$ in the target distribution. Meanwhile, let $K^{st} \in \mathbb{R}^{M \times N}$ (i.e., $(K^{st})_{ij} = \kappa(\mathbf{z}_i^s - \mathbf{z}_j^t)$) denote the Gram matrix from source distribution to target distribution for input variable $\mathbf{z}$, and $L^{st} \in \mathbb{R}^{M \times N}$ the Gram matrix from source distribution to target distribution for predicted output $\hat{y}$. Similarly, let $K^{ts} \in \mathbb{R}^{N \times M}$ (i.e., $(K^{ts})_{ij} = \kappa(\mathbf{z}_i^t - \mathbf{z}_j^s)$) denote the Gram matrix from target distribution to source distribution for input variable $\mathbf{z}$, and $L^{ts} \in \mathbb{R}^{N \times M}$ the Gram matrix from target distribution to source distribution for predicted output $y$. The empirical estimation of $D_{CCS}(p_s(y|\mathbf{z}); p_t(y|\mathbf{z}))$ is given by:*

$$
\widehat{D}_{CCS}(p_s(\hat{y}|\mathbf{z}); p_t(\hat{y}|\mathbf{z})) \approx \log\left(\sum_{j=1}^M \left(\frac{\sum_{i=1}^M K_{ji}^s L_{ji}^s}{(\sum_{i=1}^M K_{ji}^s)^2}\right)\right) + \log\left(\sum_{j=1}^N \left(\frac{\sum_{i=1}^N K_{ji}^t L_{ji}^t}{(\sum_{i=1}^N K_{ji}^t)^2}\right)\right)
$$
$$
- \log\left(\sum_{j=1}^M \left(\frac{\sum_{i=1}^N K_{ji}^{st} L_{ji}^{st}}{(\sum_{i=1}^M K_{ji}^s)(\sum_{i=1}^N K_{ji}^{st})}\right)\right) - \log\left(\sum_{j=1}^N \left(\frac{\sum_{i=1}^M K_{ji}^{ts} L_{ji}^{ts}}{(\sum_{i=1}^M K_{ji}^{ts})(\sum_{i=1}^N K_{ji}^t)}\right)\right).
\tag{72}
$$

In the following, we first demonstrate how to estimate the two conditional quadratic terms (i.e., $\int_{\mathcal{Z}} \int_{\mathcal{Y}} \frac{p_s^2(\mathbf{z},\mathbf{y})}{p_s^2(\mathbf{z})} \, \mathrm{d}\mathbf{z}\mathrm{d}\mathbf{y}$ and $\int_{\mathcal{Z}} \int_{\mathcal{Y}} \frac{p_t^2(\mathbf{z},\mathbf{y})}{p_t^2(\mathbf{z})} \, \mathrm{d}\mathbf{z}\mathrm{d}\mathbf{y}$) from samples. We then demonstrate how to estimate the cross term (i.e., $\int_{\mathcal{Z}} \int_{\mathcal{Y}} \frac{p_s(\mathbf{z},\mathbf{y})p_t(\mathbf{z},\mathbf{y})}{p_s(\mathbf{z})p_t(\mathbf{z})} \, \mathrm{d}\mathbf{z}\mathrm{d}\mathbf{y}$). We finally explain the empirical estimation of $D_{\text{CS}}(p_s(\mathbf{y}|\mathbf{z}); p_t(\mathbf{y}|\mathbf{z}))$.

*Proof.* The following proof follows directly from [Yu et al., 2023].

[The conditional quadratic term]

The empirical estimation of $\int_{\mathcal{Z}} \int_{\mathcal{Y}} \frac{p_s^2(\mathbf{z},\mathbf{y})}{p_s^2(\mathbf{z})} \, \mathrm{d}\mathbf{z}\mathrm{d}\mathbf{y}$ can be expressed as:

$$
\int_{\mathcal{Z}} \int_{\mathcal{Y}} \frac{p_s^2(\mathbf{z},\mathbf{y})}{p_s^2(\mathbf{z})} \, \mathrm{d}\mathbf{z}\mathrm{d}\mathbf{y} = \mathbb{E}_{p_s(Z,Y)}\left[\frac{p_s(Z,Y)}{p_s^2(Z)}\right] \approx \frac{1}{M}\sum_{j=1}^M \frac{p_s(\mathbf{z}_j, \mathbf{y}_j)}{p_s^2(\mathbf{z}_j)}.
\tag{73}
$$

By kernel density estimator (KDE), we have:

$$
\frac{p_s(\mathbf{z}_j, \mathbf{y}_j)}{p_s^2(\mathbf{z}_j)} \approx M \frac{\sum_{i=1}^M \kappa_\sigma(\mathbf{z}_j^{p_s} - \mathbf{z}_i^{p_s})\kappa_\sigma(\mathbf{y}_j^{p_s} - \mathbf{y}_i^{p_s})}{\left(\sum_{i=1}^M \kappa_\sigma(\mathbf{z}_j^{p_s} - \mathbf{z}_i^{p_s})\right)^2}.
\tag{74}
$$

Therefore,

$$
\int_{\mathcal{Z}} \int_{\mathcal{Y}} \frac{p_s^2(\mathbf{z},\mathbf{y})}{p_s^2(\mathbf{z})} \, \mathrm{d}\mathbf{z}\mathrm{d}\mathbf{y} \approx \sum_{j=1}^M \left(\frac{\sum_{i=1}^M \kappa_\sigma(\mathbf{z}_j^{p_s} - \mathbf{z}_i^{p_s})\kappa_\sigma(\mathbf{y}_j^{p_s} - \mathbf{y}_i^{p_s})}{\left(\sum_{i=1}^M \kappa_\sigma(\mathbf{z}_j^{p_s} - \mathbf{z}_i^{p_s})\right)^2}\right).
\tag{75}
$$

Similarly, the empirical estimation of $\int_{\mathcal{Z}} \int_{\mathcal{Y}} \frac{p_t^2(\mathbf{z},\mathbf{y})}{p_t^2(\mathbf{z})} \, \mathrm{d}\mathbf{z}\mathrm{d}\mathbf{y}$ is given by:

$$
\int_{\mathcal{Z}} \int_{\mathcal{Y}} \frac{p_t^2(\mathbf{z},\mathbf{y})}{p_t^2(\mathbf{z})} \, \mathrm{d}\mathbf{z}\mathrm{d}\mathbf{y} \approx \sum_{j=1}^N \left(\frac{\sum_{i=1}^N \kappa_\sigma(\mathbf{z}_j^{p_t} - \mathbf{z}_i^{p_t})\kappa_\sigma(\mathbf{y}_j^{p_t} - \mathbf{y}_i^{p_t})}{\left(\sum_{i=1}^N \kappa_\sigma(\mathbf{z}_j^{p_t} - \mathbf{z}_i^{p_t})\right)^2}\right).
\tag{76}
$$

[The cross term]

Again, the empirical estimation of $\int_{\mathcal{Z}} \int_{\mathcal{Y}} \frac{p_s(\mathbf{z},\mathbf{y})p_t(\mathbf{z},\mathbf{y})}{p_s(\mathbf{z})p_t(\mathbf{z})} \, \mathrm{d}\mathbf{z}\mathrm{d}\mathbf{y}$ can be expressed as:

$$
\int_{\mathcal{Z}} \int_{\mathcal{Y}} \frac{p_s(\mathbf{z},\mathbf{y})p_t(\mathbf{z},\mathbf{y})}{p_s(\mathbf{z})p_t(\mathbf{z})} \, \mathrm{d}\mathbf{z}\mathrm{d}\mathbf{y} = \mathbb{E}_{p_s(Z,Y)}\left[\frac{p_t(Z,Y)}{p_s(X)p_t(Z)}\right] \approx \frac{1}{M}\sum_{j=1}^M \frac{p_t(\mathbf{z}_j, \mathbf{y}_j)}{p_s(\mathbf{z}_j)p_t(\mathbf{z}_j)}.
\tag{77}
$$

By KDE, we further have:

$$\frac{p_t(\mathbf{z}_j, \mathbf{y}_j)}{p_s(\mathbf{z}_j)p_t(\mathbf{z}_j)} \approx M \frac{\sum_{i=1}^{N} \kappa_\sigma(\mathbf{z}_j^{p_s} - \mathbf{z}_i^{p_t})\kappa_\sigma(\mathbf{y}_j^{p_s} - \mathbf{y}_i^{p_t})}{\sum_{i=1}^{M} \kappa_\sigma(\mathbf{z}_j^{p_s} - \mathbf{z}_i^{p_s}) \sum_{i=1}^{N} \kappa_\sigma(\mathbf{z}_j^{p_s} - \mathbf{z}_i^{p_t})}. \tag{78}$$

Therefore,

$$\int_{\mathcal{Z}} \int_{\mathcal{Y}} \frac{p_s(\mathbf{z}, \mathbf{y})p_t(\mathbf{z}, \mathbf{y})}{p_s(\mathbf{z})p_t(\mathbf{z})} \, \mathrm{d}\mathbf{z}\mathrm{d}\mathbf{y} \approx \sum_{j=1}^{M} \left( \frac{\sum_{i=1}^{N} \kappa_\sigma(\mathbf{z}_j^{p_s} - \mathbf{z}_i^{p_t})\kappa_\sigma(\mathbf{y}_j^{p_s} - \mathbf{y}_i^{p_t})}{\sum_{i=1}^{M} \kappa_\sigma(\mathbf{z}_j^{p_s} - \mathbf{z}_i^{p_s}) \sum_{i=1}^{N} \kappa_\sigma(\mathbf{z}_j^{p_s} - \mathbf{z}_i^{p_t})} \right). \tag{79}$$

Note that, one can also empirically estimate $\int_{\mathcal{Z}} \int_{\mathcal{Y}} \frac{p_s(\mathbf{z}, \mathbf{y})p_t(\mathbf{z}, \mathbf{y})}{p_s(\mathbf{z})p_t(\mathbf{z})} \, \mathrm{d}x\mathrm{d}y$ over $p_t(\mathbf{z}, \mathbf{y})$, which can be expressed as:

$$\int_{\mathcal{X}} \int_{\mathcal{Y}} \frac{p_s(\mathbf{z}, \mathbf{y})p_t(\mathbf{z}, \mathbf{y})}{p_s(\mathbf{z})p_t(\mathbf{z})} \, \mathrm{d}\mathbf{z}\mathrm{d}\mathbf{y} = \mathbb{E}_{p_t(X,Y)} \left[ \frac{p_s(X,Y)}{p_s(X)p_t(X)} \right] \approx \frac{1}{N} \sum_{j=1}^{N} \frac{p_s(\mathbf{z}_j, \mathbf{y}_j)}{p_s(\mathbf{z}_j)p_t(\mathbf{z}_j)}$$

$$\approx \sum_{j=1}^{N} \left( \frac{\sum_{i=1}^{M} \kappa_\sigma(\mathbf{z}_j^{p_t} - \mathbf{z}_i^{p_s})\kappa_\sigma(\mathbf{y}_j^{p_t} - \mathbf{y}_i^{p_s})}{\sum_{i=1}^{M} \kappa_\sigma(\mathbf{z}_j^{p_t} - \mathbf{z}_i^{p_s}) \sum_{i=1}^{N} \kappa_\sigma(\mathbf{z}_j^{p_t} - \mathbf{z}_i^{p_t})} \right). \tag{80}$$

[Empirical Estimation]

Let $K^s$ and $L^s$ denote, respectively, the Gram matrices for the input variable $\mathbf{z}$ and output variable $\mathbf{y}$ in the distribution $p_s$ from the source domain. Further, let $(K)_{ji}$ denotes the $(j, i)$-th element of a matrix $K$ (i.e., the $j$-th row and $i$-th column of $K$). We have:

$$\int_{\mathcal{Z}} \int_{\mathcal{Y}} \frac{p_s^2(\mathbf{z}, \mathbf{y})}{p_s^2(\mathbf{z})} \, \mathrm{d}\mathbf{z}\mathrm{d}\mathbf{y} \approx \sum_{j=1}^{M} \left( \frac{\sum_{i=1}^{M} K_{ji}^s L_{ji}^s}{(\sum_{i=1}^{M} K_{ji}^s)^2} \right). \tag{81}$$

Similarly, let $K^t$ and $L^t$ denote, respectively, the Gram matrices for input variable $\mathbf{z}$ and output variable $\mathbf{y}$ in the distribution $p_t$ from the target domain. We have:

$$\int_{\mathcal{Z}} \int_{\mathcal{Y}} \frac{p_t^2(\mathbf{z}, \mathbf{y})}{p_t^2(\mathbf{z})} \, \mathrm{d}\mathbf{z}\mathrm{d}\mathbf{y} \approx \sum_{j=1}^{N} \left( \frac{\sum_{i=1}^{N} K_{ji}^t L_{ji}^t}{(\sum_{i=1}^{N} K_{ji}^t)^2} \right). \tag{82}$$

Further, let $K^{st} \in \mathbb{R}^{M \times N}$ denote the Gram matrix between distributions $p_s$ and $p_t$ for input variable $\mathbf{z}$, and $L^{st}$ the Gram matrix between distributions $p_s$ and $p_t$ for output variable $\mathbf{y}$. According to Eq. (79), we have:

$$\int_{\mathcal{Z}} \int_{\mathcal{Y}} \frac{p_s(\mathbf{z}, \mathbf{y})p_t(\mathbf{z}, \mathbf{y})}{p_s(\mathbf{z})p_t(\mathbf{z})} \, \mathrm{d}\mathbf{z}\mathrm{d}\mathbf{y} \approx \sum_{j=1}^{M} \left( \frac{\sum_{i=1}^{N} K_{ji}^{st} L_{ji}^{st}}{(\sum_{i=1}^{M} K_{ji}^s)(\sum_{i=1}^{N} K_{ji}^{st})} \right). \tag{83}$$

Therefore, according to Eqs. (81)-(83), an empirical estimate of $D_{\mathrm{CS}}(p_s(\mathbf{y}|\mathbf{z}); p_t(\mathbf{y}|\mathbf{z}))$ is given by:

$$D_{\mathrm{CS}}(p_s(\mathbf{y}|\mathbf{z}); p_t(\mathbf{y}|\mathbf{z})) \approx \log \left( \sum_{j=1}^{M} \left( \frac{\sum_{i=1}^{M} K_{ji}^s L_{ji}^s}{(\sum_{i=1}^{M} K_{ji}^s)^2} \right) \right) + \log \left( \sum_{j=1}^{N} \left( \frac{\sum_{i=1}^{N} K_{ji}^t L_{ji}^t}{(\sum_{i=1}^{N} K_{ji}^t)^2} \right) \right)$$

$$- 2 \log \left( \sum_{j=1}^{M} \left( \frac{\sum_{i=1}^{N} K_{ji}^{st} L_{ji}^{st}}{(\sum_{i=1}^{M} K_{ji}^s)(\sum_{i=1}^{N} K_{ji}^{st})} \right) \right). \tag{84}$$

Note that, according to Eq. (80), $D_{\mathrm{CS}}(p_s(\mathbf{y}|\mathbf{z}); p_t(\mathbf{y}|\mathbf{z}))$ can also be expressed as:

$$D_{\mathrm{CS}}(p_s(\mathbf{y}|\mathbf{z}); p_t(\mathbf{y}|\mathbf{z})) \approx \log \left( \sum_{j=1}^{M} \left( \frac{\sum_{i=1}^{M} K_{ji}^s L_{ji}^s}{(\sum_{i=1}^{M} K_{ji}^s)^2} \right) \right) + \log \left( \sum_{j=1}^{N} \left( \frac{\sum_{i=1}^{N} K_{ji}^t L_{ji}^t}{(\sum_{i=1}^{N} K_{ji}^t)^2} \right) \right)$$

$$- 2 \log \left( \sum_{j=1}^{N} \left( \frac{\sum_{i=1}^{M} K_{ji}^{ts} L_{ji}^{ts}}{(\sum_{i=1}^{M} K_{ji}^{ts})(\sum_{i=1}^{N} K_{ji}^t)} \right) \right). \tag{85}$$

Therefore, to obtain a consistent and symmetric expression, we estimate $D_{\text{CS}}(p_s(\mathbf{y}|\mathbf{z}); p_t(\mathbf{y}|\mathbf{z}))$ by:

$$D_{\text{CS}}(p_s(\mathbf{y}|\mathbf{z}); p_t(\mathbf{y}|\mathbf{z})) \approx$$
$$\log\left(\sum_{j=1}^{M}\left(\frac{\sum_{i=1}^{M} K_{ji}^s L_{ji}^s}{(\sum_{i=1}^{M} K_{ji}^s)^2}\right)\right) + \log\left(\sum_{j=1}^{N}\left(\frac{\sum_{i=1}^{N} K_{ji}^t L_{ji}^t}{(\sum_{i=1}^{N} K_{ji}^t)^2}\right)\right)$$
$$- \log\left(\sum_{j=1}^{M}\left(\frac{\sum_{i=1}^{N} K_{ji}^{st} L_{ji}^{st}}{(\sum_{i=1}^{M} K_{ji}^s)(\sum_{i=1}^{N} K_{ji}^{st})}\right)\right) - \log\left(\sum_{j=1}^{N}\left(\frac{\sum_{i=1}^{M} K_{ji}^{ts} L_{ji}^{ts}}{(\sum_{i=1}^{M} K_{ji}^{ts})(\sum_{i=1}^{N} K_{ji}^t)}\right)\right). \tag{86}$$

$\square$

# C   ADDITIONAL EXPERIMENTAL RESULTS AND DETAILS

The demo code of the proposed CS-adv in the OfficeHome data is provided in `https://anonymous.4open.science/r/CS-adv-58E5`.

## C.1   DETAILS ON CONDITIONAL DIVERGENCE TEST

The conditional KL divergence, by the chain rule, can be decomposed as:

$$D_{\text{KL}}(p^s(y|\mathbf{x}); p^t(y|\mathbf{x})) = D_{\text{KL}}(p^s(\mathbf{x}, y); p^t(\mathbf{x}, y))$$
$$- D_{\text{KL}}(p^s(\mathbf{x}); p^t(\mathbf{x})). \tag{87}$$

We estimate both terms in Eq. (87) with the $k$-NN estimator [Wang et al., 2009] ($k = 3$), due to its popularity, simplicity and effectiveness. However, we would like to emphasis here that the $k$-NN estimator itself is not differentiable, which hinders its practical usage in deep UDA.

The empirical estimation of $D_{\text{CCS}}(p^s(\hat{y}|\mathbf{x}); p^t(\hat{y}|\mathbf{x}))$ is given by:

$$\widehat{D}_{\text{CS}}(p^s(\hat{y}|\mathbf{x}); p^t(\hat{y}|\mathbf{x}))$$
$$\approx \log(\sum_{j=1}^{M}(\frac{\sum_{i=1}^{M} K_{ji}^s L_{ji}^s}{(\sum_{i=1}^{M} K_{ji}^s)^2})) + \log(\sum_{j=1}^{N}(\frac{\sum_{i=1}^{N} K_{ji}^t L_{ji}^t}{(\sum_{i=1}^{N} K_{ji}^t)^2}))$$
$$- \log(\sum_{j=1}^{M}(\frac{\sum_{i=1}^{N} K_{ji}^{st} L_{ji}^{st}}{(\sum_{i=1}^{M} K_{ji}^s)(\sum_{i=1}^{N} K_{ji}^{st})})) - \log(\sum_{j=1}^{N}(\frac{\sum_{i=1}^{M} K_{ji}^{ts} L_{ji}^{ts}}{(\sum_{i=1}^{M} K_{ji}^{ts})(\sum_{i=1}^{N} K_{ji}^t)})). \tag{88}$$

As an alternative, the conditional MMD can be estimated as [Ren et al., 2016]:

$$\widehat{D}_{\text{MMD}}(p^s(\hat{y}|\mathbf{x}); p^t(\hat{y}|\mathbf{x}))$$
$$= \text{tr}\left(L^s(\tilde{K}^s)^{-1} K^s(\tilde{K}^s)^{-1}\right) + \text{tr}\left(L^t(\tilde{K}^t)^{-1} K^t(\tilde{K}^t)^{-1}\cdot\right) - 2\cdot\text{tr}\left(L^{st}(\tilde{K}^t)^{-1} K^{ts}(\tilde{K}^s)^{-1}\cdot\right), \tag{89}$$

in which $\tilde{K} = K + \lambda I$.

Fig. 8 demonstrates the three synthetic datasets in which the set (a) and set (b) have much obvious difference in the conditional density $p(y|\mathbf{x})$; whereas the difference in set (a) and set (c) is relatively weak. Algorithm 1 summarizes the way to test the equivalence between two conditional densities.

## C.2   DETAILS ON DISTANCE METRIC MINIMIZATION

We illustrate the training scheme of our CS divergence-based distance metric minimization method in Fig. 10. For matching the latent representation $\mathbf{z}$ extracted by the feature extractor $f$, we use $D_{\text{CS}}$. For the conditional distribution $p(\hat{y}|\mathbf{z})$ alignment (classifier adaptation), we adopt $D_{\text{CCS}}$. Additionally, we use cross entropy loss $L_{\text{CE}}$ on the source domain. In the end, we train three losses jointly:

$$L = L_{\text{CE}} + \lambda D_{\text{CS}} + \beta D_{\text{CCS}}, \tag{90}$$

where $\lambda$ and $\beta$ are the weighting hyeprparameters.

**Algorithm 1:** Test for the equivalence between two conditional densities

**Input:** Two groups of observations $\psi_s = \{(\mathbf{x}_i^s, \mathbf{y}_i^s)\}_{i=1}^M$ and $\psi_t = \{(\mathbf{x}_i^t, \mathbf{y}_i^t)\}_{i=1}^N$; Permutation number $P$; Significance level $\alpha$.

**Output:** Test *decision* (Is $\mathcal{H}_0 : p_s(\mathbf{y}|\mathbf{x}) = p_t(\mathbf{y}|\mathbf{x})$ *True* or *False*?).

1: Compute conditional divergence value $d_0$ on $\psi_s$ and $\psi_t$ with one of the conditional divergence measures (e.g., conditional KL, or class conditional MMD, or conditional MMD, or conditional CS divergence).
2: **for** $m = 1$ to $P$ **do**
3:   $(\psi_s^m, \psi_t^m) \leftarrow$ random split of $\psi_s \bigcup \psi_t$.
4:   Compute conditional divergence value $d_m$ on $\psi_s^m$ and $\psi_t^m$ with the selected conditional divergence measure.
5: **end for**
6: **if** $\frac{1+\sum_{m=1}^P \mathbf{1}[d_0 \leq d_t]}{1+P} \leq \alpha$ **then**
7:   $decision \leftarrow False$
8: **else**
9:   $decision \leftarrow True$
10: **end if**
11: **return** *decision*

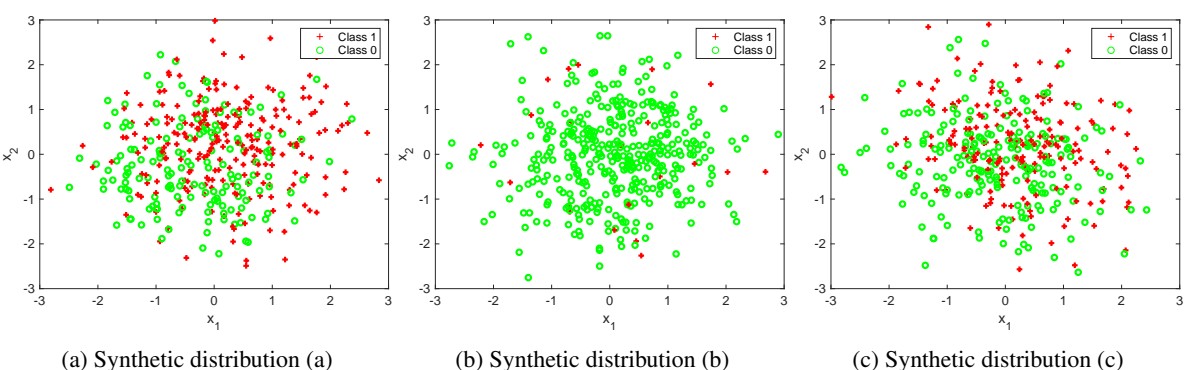

(a) Synthetic distribution (a)  (b) Synthetic distribution (b)  (c) Synthetic distribution (c)

Figure 8: Visualization of the synthetic datastes to test the power to discriminate two conditional distributions. In each plot, $x$-axis is the first dimension of $\mathbf{x}$, denote as $x_1$; $y$-axis is the second dimension of $\mathbf{x}$, denote as $x_2$. Different labels are marked with red and green, respectively.

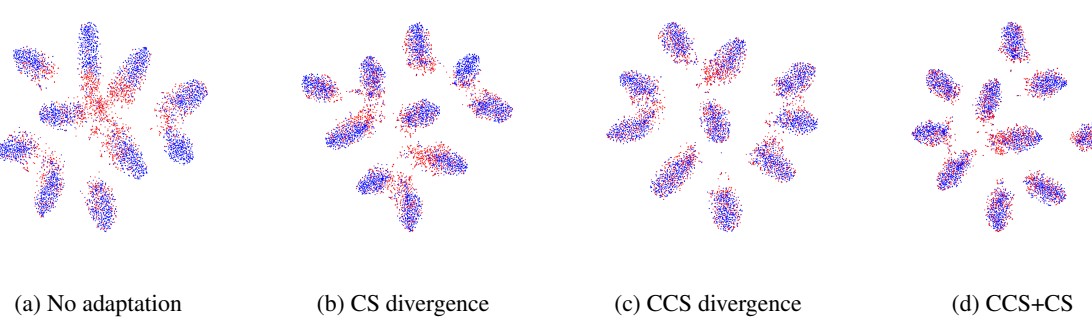

(a) No adaptation  (b) CS divergence  (c) CCS divergence  (d) CCS+CS

Figure 9: t-SNE visualization of feature trained without adaptation (9a), with CS divergence (9b, with CCS divergence (9c), and with both CCS and CS divergences (9d).

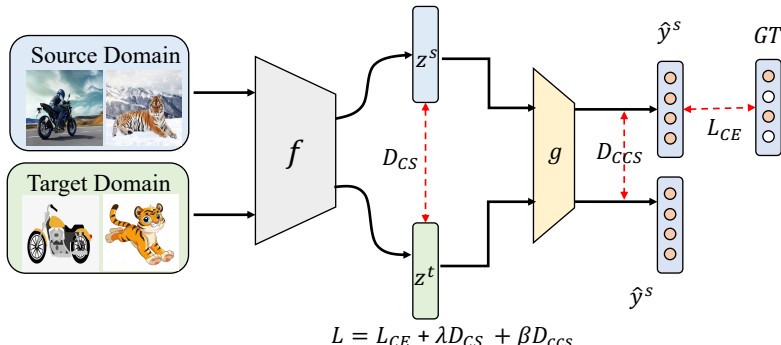

Figure 10: The diagram of distance metric minimization framework. $D_{\text{CS}}$ is used to align the latent representation $p(\mathbf{z})$, while $D_{\text{CCS}}$ matches the conditional distribution $p(\hat{y}|\mathbf{z})$

.

## C.3 COMPARISON ON DIGITS

| | M→U | U→M | Avg |
|---|---|---|---|
| DANN [Ganin et al., 2016] | 91.8 | 94.7 | 93.3 |
| CDAN [Long et al., 2018] | 93.9 | 96.9 | 95.4 |
| f-DAL [Acuna et al., 2021] | 95.3 | 97.3 | 96.3 |
| CS-adv (ours) | **95.5** | **98.1** | **96.8** |

Table 5: Accuracy on the Digits datasets.

In Table 5, we present the comparison between the proposed CS-adv method and other methods on the Digits dataset. It shows that the proposed method outperforms other methods, including f-DAL.

## C.4 ADDITIONAL ABLATION STUDY

**t-SNE visualization** In order to better understand the adaptation ability of CS and CCS divergence, we use t-SNE [Van der Maaten and Hinton, 2008] to visualize the feature trained without adaptation (Fig. 9a), with CS divergence (Fig. 9b, with CCS divergence (Fig. 9c), and with both CCS and CS divergences (Fig. 9d). The model is trained as introduced in Section 4.2 in the main text. Fig 9 shows the aligned quality on **M→U** task. As shown in Fig 9, CS divergence has a worse performance on inter-class separability, while CCS divergence can alleviate this issue. This can also be observed in Fig. 9d, where CCS divergence is added on top of CS divergence and leads to better separability compared with Fig. 9b. Hence, modeling the conditional distribution alignment is necessary and the proposed CCS divergence has an advantage.

**CCS with kSHOT** We investigate integrating the CCS divergence into kSHOT [Sun et al., 2022], an representative SOTA UDA approach. As kSHOT is based on SHOT [Liang et al., 2020] which freezes the classifier for the target domain, we only fine-tune the classifier part using CCS divergence to further enhance it by transferring the conditional distribution. The results in Table 6 show improvements on the Office-Home dataset.

| Method | Ar→Cl | Ar→Pr | Ar→Rw | Cl→Ar | Cl→Pr | Cl→Rw | Pr→Ar | Pr→Cl | Pr→Rw | Rw→Ar | Rw→Cl | Rw→Pr | Avg |
|---|---|---|---|---|---|---|---|---|---|---|---|---|---|
| kSHOT [Sun et al., 2022] | 58.2 | 80.0 | 82.9 | 71.1 | 80.3 | 80.7 | 71.3 | 56.8 | 83.2 | 75.5 | 60.3 | 86.6 | 73.9 |
| CCS+KSHOT | **58.9** | **81.6** | **83.4** | **71.3** | **81.2** | **80.8** | **71.6** | 56.5 | 82.9 | **75.8** | **61.2** | **86.7** | **74.3** |

Table 6: Compare with KSHOT [Sun et al., 2022] on **Office-Home**.

**Kernel Density Estimation visualization** In order to show that it is reasonable to use the predicted pseudo $\hat{y}$ (similar to previous papers) and the necessity of aligning both marginal and conditional distribution, we draw the Kernel Density Estimation (KDE) visualization of $p(\mathbf{z}, y)$ and $p(\mathbf{z}, \hat{y})$ in Fig. 11. We train our model on the Digits M→U task and visualize the KDE of $p(\mathbf{z}, y)$ and $p(\mathbf{z}, \hat{y})$ in the target domain (dimension reduction is performed). As the same $\mathbf{z}$ is used for both $p(\mathbf{z}, y)$ and $p(\mathbf{z}, \hat{y})$, $p(\mathbf{z}, \hat{y})$ is close to $p(\mathbf{z}, y)$ only when $p(\hat{y}|\mathbf{z})$ effectively approximates $p(y|\mathbf{z})$. In each subfigure, the

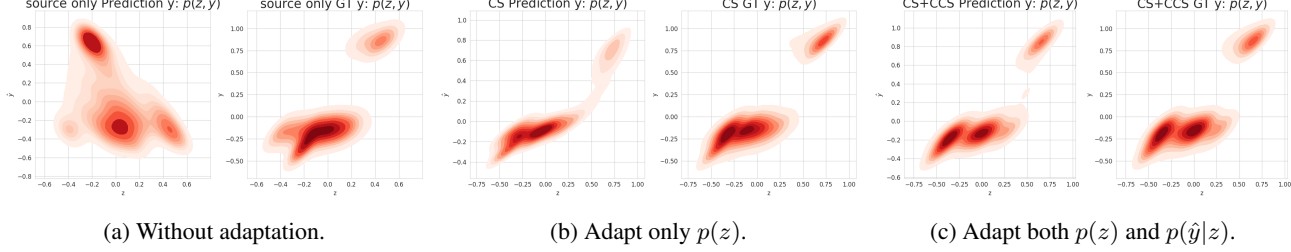

| (a) Without adaptation. | (b) Adapt only $p(z)$. | (c) Adapt both $p(z)$ and $p(\hat{y}|z)$. |

Figure 11: No adversarial training. Kernel Density Estimation (KDE) visualization of $p(\mathbf{z}, y)$ and $p(\mathbf{z}, \hat{y})$ in the target domain(after dimension reduction). $y$ and $\hat{y}$ are ground truth and predicted pseudo labels, respectively. $p(\mathbf{z}, \hat{y})$ is close to $p(\mathbf{z}, y)$ only when $p(\hat{y}|\mathbf{z})$ effectively approximates $p(y|\mathbf{z})$. Aligning $p(\mathbf{z})$ only (11b) cannot ensure the approximation of $p(\mathbf{z}, y)$, while adding conditional alignment with pseudo labels closely approximates $p(\mathbf{z}, y)$.

left shows the joint distribution $p(\mathbf{z}, \hat{y})$ from the model prediction, while the right illustrates the joint distribution with the ground truth label $p(\mathbf{z}, y)$. Fig. 11 shows aligning $p(\mathbf{z})$ only (Fig. 11b) cannot ensure the approximation of $p(\mathbf{z}, y)$, while adding conditional alignment with pseudo labels (Fig. 11c) shows a close approximation of $p(\mathbf{z}, y)$.

**Comparison on Office-Caltech-10**  In this section, we present an additional ablation study analogous to Section 4.1.2 in the main manuscript. We employ another toy dataset, Office-Caltech-10 [Fernando et al., 2014], to conduct a comprehensive comparison with both MMD and Joint Distribution MMD (JPMMD). The Office-Caltech-10 dataset comprises 10 classes with an image size of $3 \times 28 \times 28$. It includes four domains: Amazon, Webcam, Caltech, and DSLR. We selected the Webcam-to-DSLR task for demonstration. The network architecture used mirrors that in Section 4.1.2, where LeNet and two fully connected layers serve as the feature extractor and nonlinear classifier, respectively. Results are depicted in Fig. 12. Both CS and CCS methods surpass MMD and Joint Distribution MMD, with CS+CCS delivering the best performance.

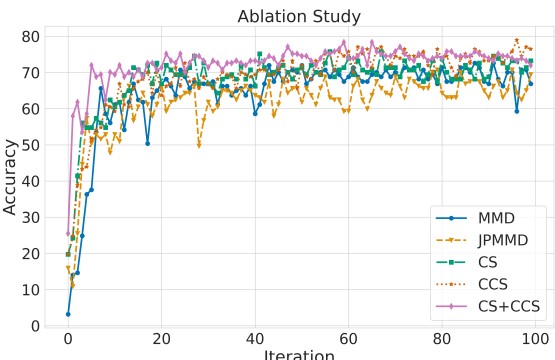

Figure 12: The ablation study of the **CS** and **CCS** components in Webcam to DSLR task, comparing with MMD and joint distribution MMD (JPMMD).

## C.5  THE EFFECT OF HYPERPARAMETERS

We conduct ablation studies on batch size and kernel size in Fig 13 on MNIST to USPS task. First, we fix the kernel size as 1 and increase the batch size. With larger batch size, the method has a better performance. Subsequently, with the batch size set at 128, we explore various kernel sizes within a specific range. It shows that our method has a stable performance with respect to kernel size in a certain range.

Additionally, in Table 7, we provide additional sensitivity analysis for $\lambda$ and $\beta$ for CS and CCS in the MNIST to USPS task. It shows that CS and CCS have stable performance for different regularization strengths. To have the same regularization strength with MMD, we keep $\lambda$ and $\beta$ as 1.

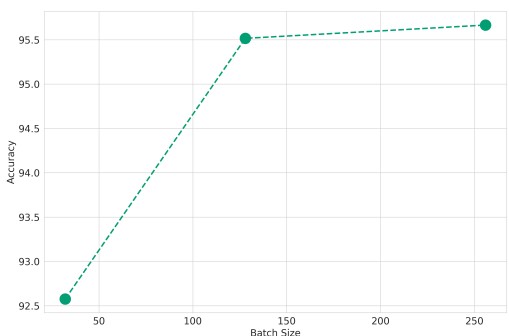

(a) The ablation study of Batch Size.

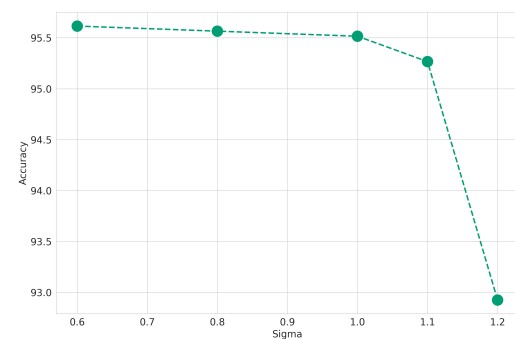

(b) The ablation study of Gaussian kernel size.

Figure 13: The ablation study of batch size and kernel size on MNIST to USPS task.

| $\lambda$ or $\beta$ | 0.1 | 0.5 | 1 | 2 | 5 | 10 |
|---|---|---|---|---|---|---|
| CS | 84.8 | 86 | 87.4 | 87.8 | 88.7 | 88.7 |
| CCS | 90.1 | 90.3 | 90.1 | 90.2 | 90.1 | 89.8 |

Table 7: Sensitivity analysis for $\lambda$ and $\beta$.