# OpenReview forum: "Domain Adaptation with Cauchy-Schwarz Divergence"
_auai.org/UAI/2024/Conference — UAI 2024 poster_

### Official Review · Reviewer_pP3L · 2024-02-29

**Q2-1 Originality-Novelty:** 3
**Q2-2 Correctness-Technical Quality:** 3
**Q2-5 Clarity Of Writing:** 4

**Q1 Summary And Contributions:**

This paper deals with the domain shift problem proposing an unsupervised approach based on the Cauchy-Schwarz (CS) Divergence. Authors provide a thoretical generalization bound based on  CS divergence and an empirical estimation of the divergence. Several experiments are performed to 1) compare the CS divergence with other metrics; 2) compare the proposed approach with SOTA methods in 3 applications.

**Q2-3 Extent To Which Claims Are Supported By Evidence:**

3: Good: the main claims are supported by convincing evidence (in the form of adequate experimental evaluation, proofs, (pseudo-)code, references, assumptions).

**Q2-4 Reproducibility:**

4: Excellent: key resources (e.g. proofs, code, data) are available and key details (e.g. proof sketches, experimental setup) are comprehensively described for competent researchers to confidently and easily reproduce the main results.

**Q3 Main Strengths:**

The work tackles the problem of DA in the challenging context of unsupervised target dataset. Authors provide a solid mathematical support for their method and conducted a large number of experiments. The method is tested on three dataset and results are compared with a large number of SOTA methods.

**Q4 Main Weakness:**

1 The notation is not helpful. Sometimes Cs and CCS are confused. CS+CCS is used for DA without adversarial training, whereas CS-adv is adopted for adversarial training. It seems that CS-adv relies only on CS instead of CS+CCS, which I suppose it is not the case.

2 The improvement over SOTA methods is modest:
- Results in Table 3 and 4 slightly improve SOTA accuracy on average
- The max accuracy obtained by f-DAL and f-DAL-CCS (Figure 3) seems to be the same.

3 Which classifier (g1, g2) did you adopt at inference? I suppose it was randomly picked. I think it would be crucial to show that, whatever classifier you pick, the performance is still the same.

**Q5 Detailed Comments To The Authors:**

In the introduction authors write “this is achieved by either using [..] divergence measures, such as […] optimal transport”. Optimal transport is a theory, whereas the metric is the Wasserstein distance.
Few lines after, authors write “The above-mentioned distance measures including MMD, KL divergence, [...]”. The divergence is not a distance.

I disagree on what reported in Sec 2 regarding OT-based models (“they align p(z) and p(y) separately and neglect their dependence”). For instance in Damodaran et al., authors define a DNN composed by a feature extractor g (providing z) and a classifier f. Eq 5 shows how g and f are simultaneously learned by minimizing the Wasserstein distance between the joint distributions p(z_s, y_s) and p(z_t, f(z_t)).

There is a typo in Sec 3.1. The probability density function of the target domain is defined as p^z instead of p^t.

Could authors clarify the sentence “The common strategy is using the discrete pseudo label from the prediction for matching the class conditional discrepancy p(z|y). However, due to the CCS divergence being able to handle the continuous variables, we use the prediction vector from the classifier yˆ = g(z) as the target label which leads to pt(yˆ|z)”

Could authors go deeper into the sentence “However, in practice, one can nevertheless control the joint divergence by minimizing the marginal and conditional counterparts separately. For simplicity, in this paper, we aim at minimizing: Eq 9”.

Please use D_CCS and D_CS consistently. In eq. 3 the conditional CS divergence is denoted by D_CS whereas in Sec 3.3 is denoted by D_CCS. Further, the abbreviation CCS is introduced multiple times in the text. Just introduce it once, at the beginning, and use CCS in the rest.

Could authors comment properly Table 1 in the manuscript? “CCS is much more powerful” is not really explanatory. For instance, in terms of rejection decisions CCS makes 3 mistakes (on the diagonal), just like class CMMD (a vs c, c vs a, c vc c).

Please report the evaluation metric related to Table 2.

**Q9 Complying With Reviewing Instructions:**

Yes

---

> ### Author Rebuttal · Authors · 2024-04-08
>
> ## Response to: "The notation is not helpful. Sometimes Cs and CCS are confused. CS+CCS is used for DA without adversarial training, whereas CS-adv is adopted for adversarial training. It seems that CS-adv relies only on CS instead of CS+CCS, which I suppose it is not the case."
>
> Thanks for the suggestion. We will carefully check the term usage of  $D_{\text{CCS}}$ and $D_{\text{CS}}$ over the whole context. Regarding CS-adv, we do use both CS and CCS divergences as shown in Eq.~14. We will make the terms clear in the revised version.
>
>
> ## Response to: "The improvement over SOTA methods is modest."
>
> We agree that the results in Tables 3 and 4 slightly improve SOTA accuracy on average as they are two widely investigated benchmarks.
> However, we would like to point out that our method performs similarly on relatively easy tasks and is better in some challenging tasks. For example, in tasks A->W, D->W, and W->D of Office-31, most baseline methods perform very well, and our method performs similarly or slightly better. However, for harder tasks, like D->A and W->A in Office-31, we clearly perform better by around 2 percent.
>
> Also, we re-run experiments of f-DAL+CCS for three runs here, and the experimental results show that CCS divergence can improve f-DAL stably.
>
> | Method    | M->U            |
> |-----------|-----------------|
> | f-DAL     | 95.4 ± 0.02     |
> | f-DAL+CCS | 95.9 ± 0.03     |
>
>
>
> ## Response to: "Which classifier (g1, g2) did you adopt at inference? It would be crucial to show that, whatever classifier you pick, the performance is still the same."
>
> Thank you for pointing out this interesting question. Following the previous bi-classifier adversarial method, MCD [Saito et al., 2018], we use the mean of the two classifiers’ outputs. However, as per request, here we show that either using one of the classifiers or using the combination, the difference is marginal.
>
> |       | $g_1$ | $g_2$ | $g_1 + g_2$ |
> |-------|-------|-------|-------------|
> | CS-adv| 95.5  | 95.6  | 95.5        |
>
>
>
> ## Response to: Terms of Optimal Transport, distance and divergence.
>
> Thank you for pointing out these less rigorous descriptions. We will change the term of Optimal Transport distance by Wasserstein distance, and carefully distinguish distance from divergence.
>
> ## Response to: "I disagree on what reported in Sec 2 regarding OT-based models, ..."
>
> We agree with the reviewer and would like to change the description as ``Typically, the transportation cost is represented as a weighted combination of costs in both feature and label spaces."
>
>
>
> ## Response to: "There is a typo in Sec 3.1. The probability density function of the target domain is defined as $p^z$ instead of $p^t$."
>
> Thanks for pointing out this typo. We will correct this to $p^t$.
>
> ## Response to: "Could authors clarify the sentence “The common strategy is using the discrete pseudo label,..., we use the prediction vector from the classifier $ \hat{y} = g(z)$ as the target label which leads to $p^t(\hat{y}|z)$”? "
>
>
>
> Sorry for the confusion. We would like to clarify that: CCS divergence is based on the kernel density estimation (KDE), which favors the estimation of continuous random variables and is less used for discrete variables. Hence, we decide to use a soft label $\hat{y}$ (the output of softmax) rather than the hard pseudo labels (after the argmax operation).
>
>
> ## Response to: "Could authors go deeper into the sentence “However, in practice, one can nevertheless control the joint divergence by minimizing the marginal and conditional counterparts separately. For simplicity, in this paper, we aim at minimizing: Eq 9”."
>
>
> In principle, we would like to minimize $D_{\text{CS}} (p^t (\mathbf{z},y); p^s(\mathbf{z},y) )$. However, Similar to TV distance and most of the f-divergences, the CS divergence does not satisfy the chain rule, that is,
> $D_{\text{CS}} (p^t (\mathbf{z},y); p^s(\mathbf{z},y) ) = D_{\text{CS}} (p^t (\mathbf{z})p^t (y|\mathbf{z}); p^s (\mathbf{z})p^s(y|\mathbf{z})) \neq D_{\text{CS}} (p^t (\mathbf{z}); p^s(\mathbf{z}) ) + D_{\text{CCS}} (p^t (y|\mathbf{z}); p^s(y|\mathbf{z}))$.
>
> However, we can still minimize $D_{\text{CS}} (p^t (\mathbf{z}); p^s(\mathbf{z}) )$ and $D_{\text{CCS}} (p^t (y|\mathbf{z}); p^s(y|\mathbf{z}))$ separately. This is because the reduction of both divergences to zero implies alignment of the joint distributions as well.
>
>
> ## Response to: "Please use $D_{\text{CCS}}$ and $D_{\text{CS}}$ consistently, ..., Just introduce it once, at the beginning, and use CCS in the rest."
>
> Thanks for the suggestion. We will carefully check the term usage of $D_{\text{CCS}}$ and $D_{\text{CS}}$ over the whole context. Also, we will ensure to introduce them once at the beginning.

---

### Official Review · Reviewer_4i6W · 2024-03-01

**Q2-1 Originality-Novelty:** 3
**Q2-2 Correctness-Technical Quality:** 3
**Q2-5 Clarity Of Writing:** 4

**Q1 Summary And Contributions:**

This paper introduces CS divergence in unsupervised domain adaptation (UDA). It shows that the CS divergence offers a tighter generalization bound on UDA than the popular KL divergence.

**Q2-3 Extent To Which Claims Are Supported By Evidence:**

4: Excellent: all claims are supported by very convincing evidence (in the form of comprehensive experimental evaluation, rigorous mathematical proofs, detailed (pseudo-)code, precise references, well-motivated and realistic assumptions) and the authors deliver what they promise.

**Q2-4 Reproducibility:**

4: Excellent: key resources (e.g. proofs, code, data) are available and key details (e.g. proof sketches, experimental setup) are comprehensively described for competent researchers to confidently and easily reproduce the main results.

**Q3 Main Strengths:**

- This paper proposes two new method CS+CCS and CS-adv which achieve state-of-the-art performance in several real datasets.
- This paper provides a theoretical result that supports the proposed method. This result says  that the CS divergence offers a tighter generalization bound on UDA than the popular KL divergence.
- This paper is clearly written.

**Q4 Main Weakness:**

- The novelty may be limited, since it mainly introduces another divergence into the UDA task.

**Q5 Detailed Comments To The Authors:**

1. There is a missing 'and' in the sentence right before equation (14).

**Q9 Complying With Reviewing Instructions:**

Yes

---

> ### Author Rebuttal · Authors · 2024-04-08
>
> **We thank Reviewer 4i6W for the comments.**
>
> ## Response to: "The novelty may be limited, since it mainly introduces another divergence into the UDA task."
>
> We would like to emphasize that a reliable divergence measure plays a pivotal role in the design of modern domain adaptation methods. Various divergences, such as total variation (TV) distance, Wasserstein distance, and Maximum Mean Discrepancy (MMD), have been proposed previously. While MMD is easy to implement and offers a closed-form estimator, it often suffers from a formal generalization error bound. On the other hand, TV distance or KL divergence facilitate generalization analysis but can be challenging to accurately estimate. Our proposed CS divergence combines the advantages of both: it features a closed-form estimator and induces a generalization error bound that is tighter than the use of KL divergence, although we acknowledge that TV distance may yield an even tighter bound than ours.
>
> Meanwhile, our extensive experiments indicate that our CS divergence outperforms Wasserstein distance-based approaches, such as ([Damodaran et al., 2018], [Fatras et al., 2021]).
>
> Furthermore, we would like to emphasize that our method provides a novel approach to align conditional distributions, which significantly surpasses conditional MMD or class conditional MMD.
>
> Finally, we would like to emphasize our paper sheds light on relevant tasks such as domain generalization or few-shot learning.
>
>
>
> ## Response to: "There is a missing 'and' in the sentence right before equation (14)."
>
> Thanks for pointing out this typo. We will correct it accordingly in the revised version.

---

### Official Review · Reviewer_qiH3 · 2024-03-08

**Q2-1 Originality-Novelty:** 3
**Q2-2 Correctness-Technical Quality:** 3
**Q2-5 Clarity Of Writing:** 4

**Q1 Summary And Contributions:**

The paper proposes Cauchy-Schwarz (CS) divergence for unsupervised domain adaptation, where this divergence is minimized in order to obtain domain-invariant latent representations and to ensure that the classifiers are consistent. The authors utilize this loss function both as a regularization term in classical training as well as in an adversarial training scheme. The proposed methods are empirically evaluated on several datasets, illustrating the advantages of CS divergence.

**Q2-3 Extent To Which Claims Are Supported By Evidence:**

3: Good: the main claims are supported by convincing evidence (in the form of adequate experimental evaluation, proofs, (pseudo-)code, references, assumptions).

**Q2-4 Reproducibility:**

3: Good: key resources (e.g. proofs, code, data) are available and key details (e.g. proofs, experimental setup) are sufficiently well-described for competent researchers to confidently reproduce the main results.

**Q3 Main Strengths:**

The paper is very well written and quite accessible. The introduction is exemplary. The mathematical notation is rigorous and well-chosen. Section 4.1.1 nicely illustrates that conditional CS is more powerful in detecting differences between conditional distributions than MMD or Kullback-Leibler divergence, and the remaining sections 4.1.2, 4.2, and 4.3 show that the proposed divergence measures can be implemented efficiently (i.e., estimation from data is not problematic due to kernel methods) and effectively (i.e., outperforming other divergence measures and/or improving existing techniques when they are supplemented by CS regularization). Also Section 3.2 is quite interesting, situating CS divergence within the existing literature on generalization bounds.

**Q4 Main Weakness:**

The paper has only few weaknesses that I will list here:
- The authors claim that they prove a tighter generalization error bound using CS divergence. This tighter bound is not explicitly stated, in my opinion. The fact that $D_{CS}\le D_{KL}$ alone (see Remark 1) is not sufficient to claim a tighter bound, especially since in (4) there is an additional factor of $1/\sqrt{2}$ compared to (9). The claim that there is a tighter bound certainly requires a more rigorous statement.
- While the experiments are well-executed, it is not clear how the hyperparameters  $\lambda$ and $\beta$ are selected (except in Sec. 4.2) or how they should be selected. The sensitivity analysis in the appendix does not cover all hyperparameters. This is a critical weakness of the experimental section.
- When reporting results for multiple domain shifts as in Tables 3 and 4, it does not make sense to report average accuracies. Indeed, if the difficulty of certain tasks are different, then the averages can be dominated by only few tasks. Therefore it is more reasonable to present average ranks for the individual methods.
- Some paragraphs are not fully clear (see below), but I expect that this can be repaired easily during the rebuttal phase.

**Q5 Detailed Comments To The Authors:**

- In Sec. 3.1, the density function for the target domain is denoted as $p^z$ -- should it not be $p^t$?
- The last sentence in Sec. 3.1 is not clear, also because the abbreviation CCS is not introduced before.
- The first sentence in Sec. 3.3 is not clear ("let us denote the predicted class probabilities for [...] are respectively [...]")
- In Proposition 4, the concept of Gram matrices must be introduced, and the kernel function  $\kappa$ is used differently than in Proposition 3 (one vs. two arguments).
- In the sentence starting with "As shown in Fig. 1", a bracket is missing.
- In Table 2, why are both the second and the third row bold?
- In Table 4, for $D\to W$, KL actually outperforms CS-adv and f-DAL. Please correct the bold indicators.
- In Fig. 2 and 3,it is not clear for how many random runs (if at all) the results are shown. Also, why does Table 4 report standard deviations, but Table 3 does not?

**Q9 Complying With Reviewing Instructions:**

Yes

---

> ### Author Rebuttal · Authors · 2024-04-08
>
> ## Response to:  "The authors claim that they prove a tighter generalization error bound using CS divergence. ..., The claim that there is a tighter bound certainly requires a more rigorous statement."
>
> We apologize for the confusion caused. In this response, we articulate bounds for general distributions more rigorously.
>
> As shown in Proposition 5, the inequality $C_1 \left[D_{\mathrm{CS}}(p;q) - \log{|K|} + 2\log C_2 \right] \leq D_{\mathrm{KL}}(p;q)$ holds for any density functions $p$ and $q$, where $C_1 =  \int_K p(\mathbf{x})\,\mathrm{d} \mathbf{x} \approx 1$ and $C_2 = C_1  \left(\int_K p^2(\mathbf{x})\,\mathrm{d} \mathbf{x} \int_K q^2(\mathbf{x})\,\mathrm{d} \mathbf{x} \right)^{-1/4} $ are two constants. Furthermore, according to Proposition 6, if $p$ and $q$ do not sufficiently overlap, as ensured by the condition $\int p^2(\mathbf{x})\, \mathrm{d} \mathbf{x} \int q^2(\mathbf{x})\, \mathrm{d} \mathbf{x} \geq \exp(2) \left(2\epsilon+T_{\cal A_\epsilon^{\complement}}\left|\cal A_\epsilon^{\complement}\right|\right)^2$ for some $\epsilon>0$, then $D_{\mathrm{TV}}\leq \sqrt{D_{\mathrm{CS}}}$. By combining these results, we have the following general error bound:
> \begin{align*}
> l_{\text{test}}
> & \leq l_{\text{train}} + M  D_{\text{TV}} (p^t (\mathbf{z},y); p^s(\mathbf{z},y)) \\\
> & \leq l_{\text{train}} + M \sqrt{ D_{\text{CS}} (p^t (\mathbf{z},y); p^s(\mathbf{z},y)) } \\\
> & \leq l_{\text{train}} + M \sqrt{ C_1^{-1} D_{\text{KL}} (p^t (\mathbf{z},y); p^s(\mathbf{z},y)) + \log{|K|} - 2\log C_2},
> \end{align*}
> provided $\int p^2(\mathbf{x})\ \mathrm{d} \mathbf{x} \int q^2(\mathbf{x})\ \mathrm{d} \mathbf{x} \geq \exp(2) \left(2\epsilon+T_{\cal A_\epsilon^{\complement}}\left|\cal A_\epsilon^{\complement}\right|\right)^2$ for some $\epsilon>0$. As discussed in Remark 5, it appears quite feasible to satisfy this condition by setting a small value for $\epsilon>0$. Moreover, the notation $\lesssim$ is no longer employed in this general error bound, and the tighter bound is explicitly given.
>
>
> We would like to justify that $C_1$ is likely to be $1$ as discussed in Proposition 5. The essence of this general bound is that the use of CS divergence is prone to lead to a tighter bound than the use of KL divergence.
> As for the factor of $1/\sqrt{2}$, as $M$ is a constant, one can use $M’ = M/\sqrt{2}$ as a new constant, which does not affect the conclusion.
>
>
>
> ## Response to: "While the experiments are well-executed, it is not clear how the hyperparameters $\lambda$ and $\beta$ are selected, ..., This is a critical weakness of the experimental section."
>
> For $\lambda$ and $\beta$, we simply use the selected hyperparameters selected from the metric-learning experiments. In the metric-learning experiment (for comparing with MMD), we use $\lambda$ and $\beta$ from {0.1, 1, 5, 10}, and found the $\lambda=1$ and $\beta=1$ give the best performance.
>
> Here, we provide additional sensitivity analysis for $\lambda$ and $\beta$ for CS and CCS in the MNIST to USPS task. It shows that CS and CCS have stable performance for different regularization strengths. To have the same regularization strength with MMD, we keep $\lambda$ and $\beta$ as 1.
>
> | $\lambda$ or $\beta$ | 0.1  | 0.5  | 1    | 2    | 5    | 10   |
> |----------------------|------|------|------|------|------|------|
> | CS                   | 84.8 | 86   | 87.4 | 87.8 | 88.7 | 88.7 |
> | CCS                  | 90.1 | 90.3 | 90.1 | 90.2 | 90.1 | 89.8 |

---

### Official Review · Reviewer_zCYg · 2024-03-18

**Q2-1 Originality-Novelty:** 2
**Q2-2 Correctness-Technical Quality:** 2
**Q2-5 Clarity Of Writing:** 2

**Q1 Summary And Contributions:**

The authors propose the use of the Cauchy-Schwarz (CS) divergence to learn representations for unsupervised domain adaptation. This is backed up by an analysis relating the CS divergence to the total variation (TV) distance and the KL divergence, arguing that CS divergence is sandwiched between the other two, benefiting from larger discriminative power than TV and a tighter bound than KL. Based on this observations, the authors propose regularizing representation learning algorithms by minimising the CS divergence of the marginal distribution of embeddings and the estimated conditional distribution of labels given embeddings. This strategy is evaluated and compared to benchmarks on a well-established benchmarks data sets.

**Q2-3 Extent To Which Claims Are Supported By Evidence:**

2: Fair: the main claims are somewhat supported by evidence (but the experimental evaluation may be weak, or does not match entirely with the claims, important baselines may be missing, proofs contain important ideas but lack rigor, algorithmic details are only discussed superficially, references are imprecise, assumptions are not sufficiently motivated or explicated, etc.).

**Q2-4 Reproducibility:**

3: Good: key resources (e.g. proofs, code, data) are available and key details (e.g. proofs, experimental setup) are sufficiently well-described for competent researchers to confidently reproduce the main results.

**Q3 Main Strengths:**

* The problem is well-motivated: UDA algorithms are notorious for making assumptions that are not satisfied by real-world problems and algorithms. This is worth studying.
* The beginning of the paper is well laid out.
* The method is compared to several algorithms on known benchmarks.

**Q4 Main Weakness:**

* There is a misalignment between the motivation (which is good) and what the paper actually produces as contributions.
* The authors state that a contribution is that "the CS divergence enables a tighter generalisation bound than the popular KL divergence". I have several objections to this claim and the reasoning that underpins it (see below).
* The empirical estimator (Figure 1) includes many components (e.g., bi-classifier) which are unrelated to the CS divergence, which makes credit assignment difficult in comparison with baseline methods.

**Q5 Detailed Comments To The Authors:**

* Misalignment between the motivation and contribution

1. One of the main points in the introduction is that several other works assume that $p_t(y | z) = p_s(y | z)$ for a learned representation $z$. However, the (approximate) bounds in (8) are not measurable in UDA since the target distribution of labels is unknown. In the empirical estimator of Proposition 4, the label distribution is substituted by the **predicted** label distribution. Clearly, this may suffer from the same issues that plague other representation learning UDA methods, which is that a representation may maintain all necessary information about labels in the source domain but lose information about labels in the target domain---especially when the two domains are far apart, which is where Propositions 1 & 2 are valid. The authors give no argument for why aligning the estimates is a valid strategy.
2. The empirical methodology includes many

* Claim that CS leads to tighter bounds and that this may be useful

1. The authors refer to the proof from Nguyen et al., (2021b) which uses the fact that $D_{TV} \leq \sqrt{\frac{1}{2}D_{KL}}$ and then argue that $\sqrt{D_{CS}}$ is sandwiched between $D_{TV}$ and $\sqrt{D_{KL}}$. However, in the main paper this is only shown to hold in the case of two Normal distributions. Given only this (we'll get to results in the appendix later), for non-Normal distributions, substituting the $D_{CS}$ for $D_{KL}$ does not result in a bound on the test error at all. No formal bound on the test error is presented, but the full argument is only given in **Remark 1**, in which they use the notation $\lesssim$ to indicate "less than or similar to", without further explanation. The proof relies on establishing conditions under which $D_{CS} \geq 1 \geq D_{TV}$, which amounts to having distributions that are sufficiently separated (lower bound on mean distance). This means that the assumptions for the result are violated when the distributions are close. This is a problem, since the goal of the algorithm is to **minimise** $D_{CS}$.

2. In Appendix A, Proposition 1 & 2 are extended to the non-Normal case, where it is shown that $D_{KL} \geq C_1[D_{CS} - \log |K| + 2\log C_2]. Depending on the sizes of the constants, this may or may not yield a valid bound. A similar argument is made for the $D_{TV}$ part. I would have liked to see a much more extensive discussion in the main paper about how this impacts the use of the bound for the representation learning algorithms used. For example, how do the conditions of Proposition 5 & 6 relate? Are they compatible?

3. The total variation distance, $D_{TV}$ itself yields a tighter bound than $D_{CS}$ but the authors argue in Remark 1 that, when $D_{TV}$ reaches its upper bound of 1, it can no longer distinguish sufficiently distinct pairs $(p,q)$. If this is the case, why is having a tighter bound than the one given by $D_{KL}$ preferable? Surely, if that argument holds, the tightest bound should be the best (and $D_{TV}$ yields a tighter bound than $D_{CS}$)? At the very least, tightness is not the reason we prefer one over the other. Generally, the chosen strategy will yield bounds on the test error that are vacuous in the domain where the proofs hold.

4. The KL divergence is often discarded as an option in UDA or other representation learning, precisely because it blows up when the two distributions don't share support. In the example of Figure 1, support overlap is clearly violated. Calling KL popular for such problems is misleading, and being tighter than KL in this case is not an impressive feat.

* It is not clear from the empirical results whether the benefits observed in UDA are due to the CS divergence.
1. Do all baselines use the same architecture and training strategy? Are they all using the bi-classifier adversarial training?
2. In Table 2, there is only a comparison with KL, but I would argue that KL is not widely used as a discrepancy/divergence metric for learning invariant representations. It would be appropriate to compare also with MMD or adversarial metrics.
3. The overall accuracy numbers, the numbers in Table 2, and the plots in Figure 2 are missing uncertainty estimates.
4. How were hyper parameters selected for baseline methods? For example, in Figure 2, was the same strength of discrepancy regularisation used for all methods? If so, is it not a problem that the MMD has a different scale? If not, how were the values chosen?

### Minor points

* $\log \hat{p}(y | z)$, and therefore $M$ may be very very large for good classifiers when there are many classes.
* The proof of Proposition 4 follows directly from Yu et al., (2023), but this is stated only in the appendix.
* The authors refer to "ablation studies" in several places, but it is not clear what is being ablated (removed)
* What is the x-axis in Figure 2? Training iterations?

**Q9 Complying With Reviewing Instructions:**

Yes

---

> ### Author Rebuttal · Authors · 2024-04-08
>
> # Summary of rebuttal
>
> Firstly, we appreciate your careful review of our submission and the insightful questions you have raised. Regarding the three main weaknesses that you listed, we would like to clarify:
>
> ## 1. Misalignment between the motivation (which is good) and what the paper actually produces as contributions.
>
> We expect that our response to your detailed comments can address your concern.
>
> ## 2. Objections to the tighter bound claim.
>
> In the following response to detailed comments, we provide a generalization error bound without the notation $\lesssim$ for comparison with the KL bound. The new bound is essentially a re-formulation of our propositions but is expressed more formally. In addition, we justify the compatibility of the conditions outlined in these propositions, particularly when assessing high-dimensional image data.
>
> ## 3. The empirical estimator (Figure 1) includes many components (e.g., bi-classifier) which are unrelated to the CS divergence, which makes credit assignment difficult in comparison with baseline methods.
>
> We are sorry for the confusion. The experimental results (Figure 2 and Table 2) reported in Section 4.1.2 are achieved only by replacing MMD (or KL divergence) with our CS or conditional CS using exactly the same architecture and a careful hyperparameter tuning, **without ANY** adversarial training or bi-classifier strategy. Following your request, we have included additional comparisons including MMD+CCS, MMD+CMMD in the rebuttal. We also report more evaluations on challenging tasks (Webcam to DSLR and Amazon to DSLR) from the Office-Caltech-10 dataset. All these results are achieved with architectures shown in Figure 9 in the Appendix, rather than Figure 1.
>
> Figure 1 is an example to show that our divergence is flexible and compatible with the modern deep learning architecture to achieve higher classification accuracy, which is comparable to state-of-the-art UDA methods that usually rely on complicated training strategies (such as [Acuna et al., 2021], [Fatras et al., 2021]). In this rebuttal, we also demonstrated that the conditional MMD may **NOT** be compatible with these modern strategies.
>
> # Summary of Contributions
>
> Before addressing your detailed comments one by one, we would like to reaffirm our main contributions:
>
> 1. Introduction of a new divergence measure to UDA, which can be used as an alternative to MMD. Moreover, conditional CS divergence works much better than conditional MMD (this point is further justified in our rebuttal).
>
> 2. Theoretical justification for choosing CS divergence by providing a generalization error bound. To our knowledge, such analysis is usually neglected in previous MMD-based UDA literature. We admit that the total variation (TV) distance may lead to a tighter bound than our bound. However, TV distance is hard to estimate accurately. Additionally, the extension of the TV distance to conditional TV distance has not been addressed.
>
> 3. Using the same architecture, same training strategy, and careful hyperparameter tuning, our divergence outperforms MMD-based and KL divergence-based methods **without ANY** adversarial training or bi-classifier strategy. In addition, our conditional CS is also compatible with modern deep learning frameworks and modern training strategies such as the bi-classifier. These strategies can further boost our performance, but may not be helpful to MMD.

---

### Meta-Review · Area_Chair_Zm4k · 2024-04-22

The authors propose using CS divergence in learning representations for unsupervised domain adaptation.
They support this proposal with an analysis that positions the CS divergence between the total variation (TV) distance and the KL divergence, highlighting its superior discriminative power relative to TV and a tighter bound compared to KL. Moreover, CS divergence is used in the loss functions as a regularization term in classical training and even in the adversarial training scheme. Experimental results demonstrate the advantages of CS divergence.

Reviewers enjoyed the clear motivation of the work and the clarity of the presentation in Introduction. However, some reviewers expressed concerns about a gap between the stated motivation and the actual contributions, and criticized inadequate comparative analysis. In their response, the authors nicely provided additional justifications and expanded experimental comparisons involving the MMD.